

# Mapping Accessibility for Earthquake Hazard Response in the Historic Urban Center of Bucharest

Cristina Merciu[1], Ioan Ianoş[1], George-Laurenţiu Merciu[2], Roy Jones[3], and George Pomeroy[4]

[1]Interdisciplinary Center of Advanced Research on Territorial Dynamics, University of Bucharest, Blvd. Regina Elisabeta, 4-12, code 030018, Romania
[2]Faculty of Geography, University of Bucharest, Blvd. Nicolae Bălcescu, 1, code 030018, Romania
[3]Geography Discipline Group, Curtin University, Perth, Western Australia 6845, Australia
[4]Geography – Earth Science Department, Shippensburg University of Pennsylvania. 1871 Old Main Drive, Shippensburg PA 17257

*Correspondence to*: Ioan Ianoş (ianos50@yahoo.com)

**Abstract.** Planning for post-disaster accessibility is essential for the provision of emergency and other services to protect life and property in impacted areas. Such planning is particularly important in congested historic districts where narrow streets and at-risk structures are more common and may even prevail. Indeed, a standard method of measuring accessibility, through the use of isochrones, may be particularly inappropriate in these congested historic areas. Bucharest, Romania, is a city with a core of historic buildings and narrow streets. Furthermore, Bucharest ranks second only to Istanbul among large European cities in terms of its seismic risk. This paper provides an accessibility simulation for central Bucharest using mapping and GIS technologies. It hypothesizes that all buildings in the Risk 1 class would collapse in an earthquake of a similar magnitude to those of 1940 and 1977. The authors then simulate accessibility impacts in the historic center of Bucharest, such as the isolation of certain areas, and blockages of some street sections. In this simulation, accessibility will be substantially compromised by anticipated and extensive building collapse. Therefore, policy makers and planners need to fully understand and incorporate the serious implications of this compromised accessibility when planning emergency services and disaster recovery responses.

## 1 Introduction

A longitudinal analysis of natural hazards in major urban areas shows an increasing awareness of the frequency of disasters and especially of earthquakes (Eshghi & Larson, 2008; Armaş, 2012; Lu & Xu, 2014). Indeed, earthquakes are among the natural disasters that generate the greatest human and material losses (Geis, 2000;



Armaş & Avram, 2008; Atanasiu & Toma, 2012). Their impacts demand a prompt response from decision makers and the wider population, through the proper management of emergency situations (Waugh & Streib, 2006).

Many areas of high seismic risk are urbanized and densely populated (Pollino et al., 2012; Vatseva et al., 2013). In addition, and coincidentally, many countries experiencing economic transitions are characterized by
urban growth that is uncontrolled and, in large and medium-sized urban centers, such growth can be especially chaotic (Salvati, 2014). Thus, an increase in the human and economic cost of such disasters can be reasonably anticipated. Furthermore, many new buildings, new structures and, sometimes, newer pieces of infrastructure frequently fail to comply with the construction regulations established for areas of differing seismic vulnerability, especially when there are strong pressures for rapid development. Finally, the characteristically
long time lags between pairs of strong earthquakes (Schweier & Markus, 2009) can dull public awareness of the potential impacts of such disasters, and render those in charge of emergency management complacent.

Earthquakes require a specific disaster planning approach (Armaş, 2008; Boştenaru Dan & Armaş, 2015). This is because, unlike disasters that can be anticipated in the short term (such as storms), there is little or no delay between the occurrence of the earthquake and the subsequent loss of life and property damage.  Therefore,
emergency response activities must be executed very quickly and efficiently (Wegscheider et al., 2013). For cities with a high earthquake risk, an important factor is public awareness of such events. This conditions the population towards the importance of quick response measures, which can help to reduce property damage and, more importantly, the number of casualties (Armaş & Avram, 2008). However, no matter how well organized the mitigation process, the disastrous effects of major earthquakes cannot be totally avoided (Momani & Salmi,
50    2012).

In recent years, seismic risk management has been more fully studied and developed so as to establish a series of priorities related to the rehabilitation of those buildings considered to be of major importance, including schools (Crowley et al., 2008; Raffaelle et al., 2013; Panahi et al., 2013), public institutions, historic buildings, and monuments (Grasso & Maugeri, 2009; Pessina & Meroni, 2009). Urban earthquake planning therefore
needs to be more proactive (Boştenaru Dan et al., 2014) and there is a demonstrated requirement for coherent urban policies (Ianoş et al., 2017) to mitigate the inevitable occurrence of blockage points during emergency interventions.

In emergency situations, the key response element is rapid accessibility to places where possible casualties may be located. Timely intervention within the first two hours is critical in saving the wounded and in
identifying the safest access routes for specific emergency equipment.



In general, natural hazard management includes the development of impact scenarios before the actual disasters occur (Bakillah et al., 2013). In this context, GIS techniques may be particularly useful in developing decision-making and response scenarios for potential earthquake disasters.

Our study shows that special attention should be paid to accessibility in the historic centers of large cities (Ianoş & Cepoiu, 2009). Historic city centers are characterized by intense pedestrian traffic and by a high proportion of attraction points (clubs, restaurants, hotels etc.) which result in high concentrations of people. Since the core of the historic center of Bucharest is characterized by a high number of buildings that were strongly affected by earthquakes in the last century, we can reasonably speculate that determining their accessibility in an emergency situation will facilitate quick intervention in areas where injured people, either direct casualties or victims of earthquake-related phenomena such as fires, gas accumulations or local flooding, are likely to be concentrated. The main objective of the study is to integrate geospatial data using thematic mapping products with GIS techniques in order to provide seismic risk management solutions for Bucharest. We therefore seek to provide, concrete data and comprehensible information that can enable decision-makers to implement and prioritize their disaster management strategies. A similar study, based on different hazard scenarios and a deep analysis on social vulnerability in Bucharest, identifies the importance of fire stations, hospitals and parks in post-disaster situations (Armaş et al., 2016).

Unlike most studies of community response following an earthquake occurrence and the critical analysis of the emergency situations management generated thereby (Pollino et al., 2012; Wegscheider et al., 2013; Lu & Xu, 2014), the present study demonstrates the importance of GIS analyses in detecting potential congestion and inaccessibility issues in areas where buildings are most likely to collapse and accessibility issues are most likely to arise as a result of an earthquake.

## 2 Case Study

Bucharest is Romania's largest city (with over two million inhabitants), the national capital, and one of the great metropoles of the Southeastern Europe (GROSEE-Espon project, 2014). Its urban evolution has been very rapid, largely occurring from the second half of the 19th century. Currently, the city occupies an area of 228 square kilometers, and possesses a housing stock predominantly consisting of multifamily apartment buildings, built during the communist period (Ianoş et al., 2016). Located about 135 km from the epicenter of the Vrancea seismic area (Lungu et al., 2000), in close proximity to the Southern Carpathian Mountains and at the junction of the Eastern European, intra-Alpine and Moesia plates (Mărmureanu et al., 2011), the city is extremely


vulnerable to earthquakes. Indeed, in a classification of European metropolitan areas with respect to potential
loss of life and damage to property, Bucharest is ranked second after Istanbul (Bala, 2014).

The historical record of Bucharest is replete with accounts of damaging earthquakes ever since the city's
foundation (Tatevossian & Albini, 2010). The Vrancea seismic area is responsible for the highest seismic risk in
Romania (Pavel et al., 2014; Ardeleanu et al., 2005). Over the past 76 years, Bucharest has been affected by
four earthquakes with a magnitude of between 6.9 and 7.7 on the Richter scale (November 1940, March 1977,
August 1986 and May 1990).

The study area for this paper is confined to central Bucharest, an area of approximately 8.33 square kilometers
(Fig. 1). The oldest part of the city is situated in the south of this area, which comprises the historic center (from
the 16$^{th}$ and 17th centuries) and the central and northern parts dating from the 18th and19th centuries. All four
earthquakes mentioned above have impacted this case study area, with the most powerful being the earthquakes
of 1940 and 1977. This sequence of earthquakes has had a cumulative effect, which explains the relative lack of
buildings dating back more than 200 years.

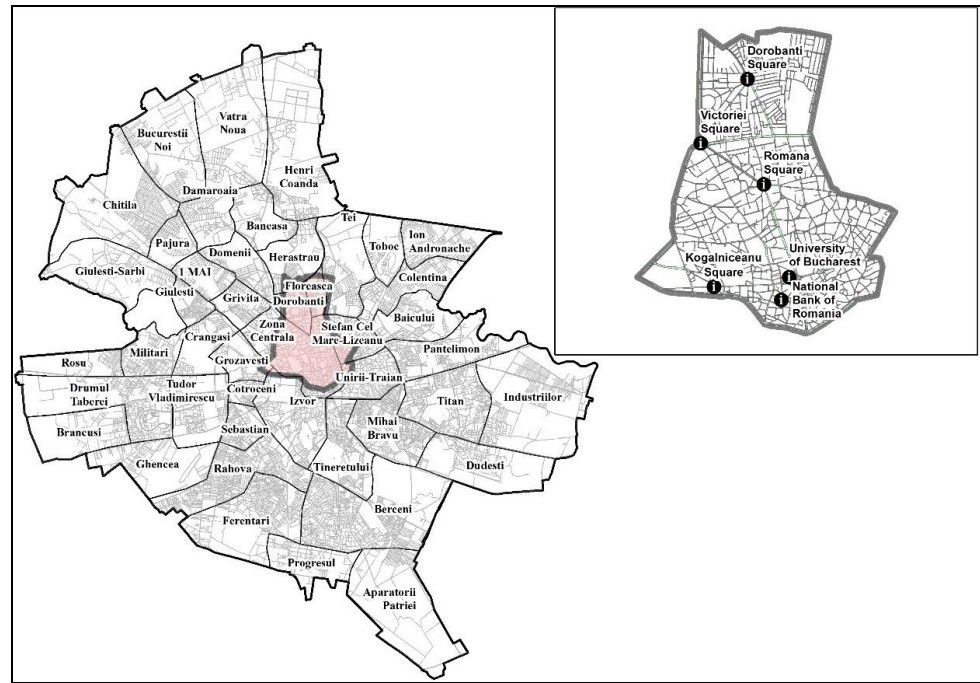

**Figure 1. Bucharest city and its surrounds**

The most serious problem is presented by the large number of buildings from the late 19th century within the
historic center, which has structurally degenerated over time and no longer meets the current building standards



with regard to assessment of the ground motion levels for the Vrancea (Romania). Not only does Bucharest have a high level of exposure to earthquake hazards, it also suffers from poorly organized civil protection services and a low level of public awareness and education concerning these seismic risks (Armaş, 2006). Nevertheless,

anticipation and anxiety are building, given the length of time that has passed since the 1977 major earthquake. In essence, there is a fear that the city will be no better prepared than it was in 1940 (Fig. 2a) or in 1977 (Fig. 2b). These figures show only a slight improvement in the standard of the disaster measures between the two dates and there is a growing recognition that greater levels of preparedness are needed.

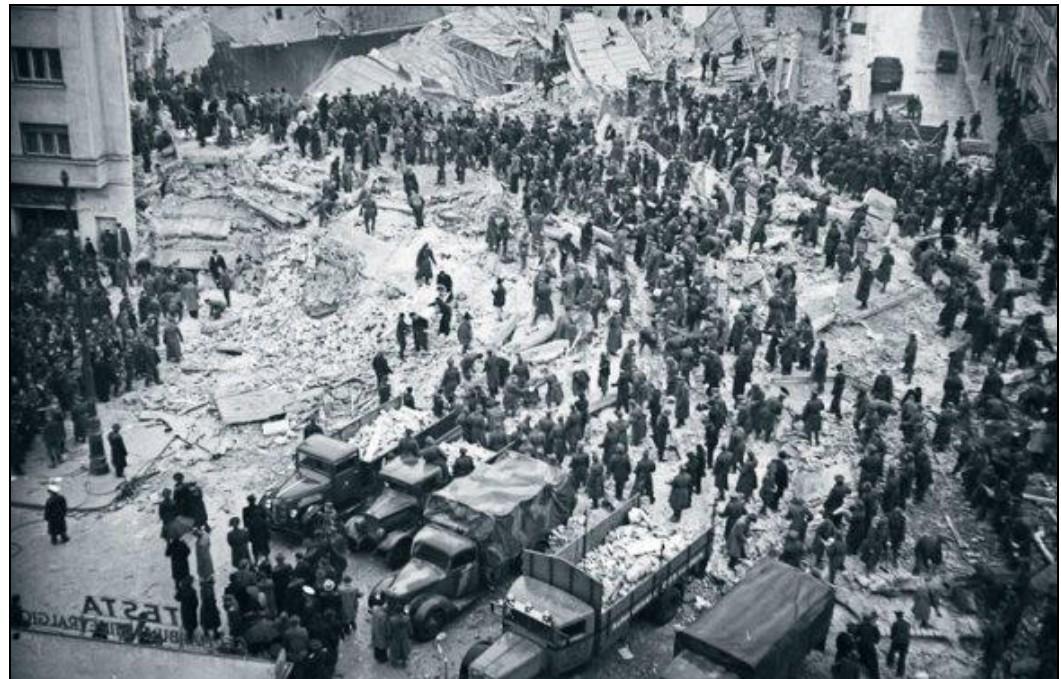

**Figure 2a. The collapse of the Carlton block in Bucharest in 1940.**

Source:

https://ro.wikipedia.org/wiki/List%C4%83_de_cutremure_%C3%AEn_Rom%C3%A2nia#/media/File:Iosif_
Berman_-_Marele_cutremur_din_anul_1940.jpg




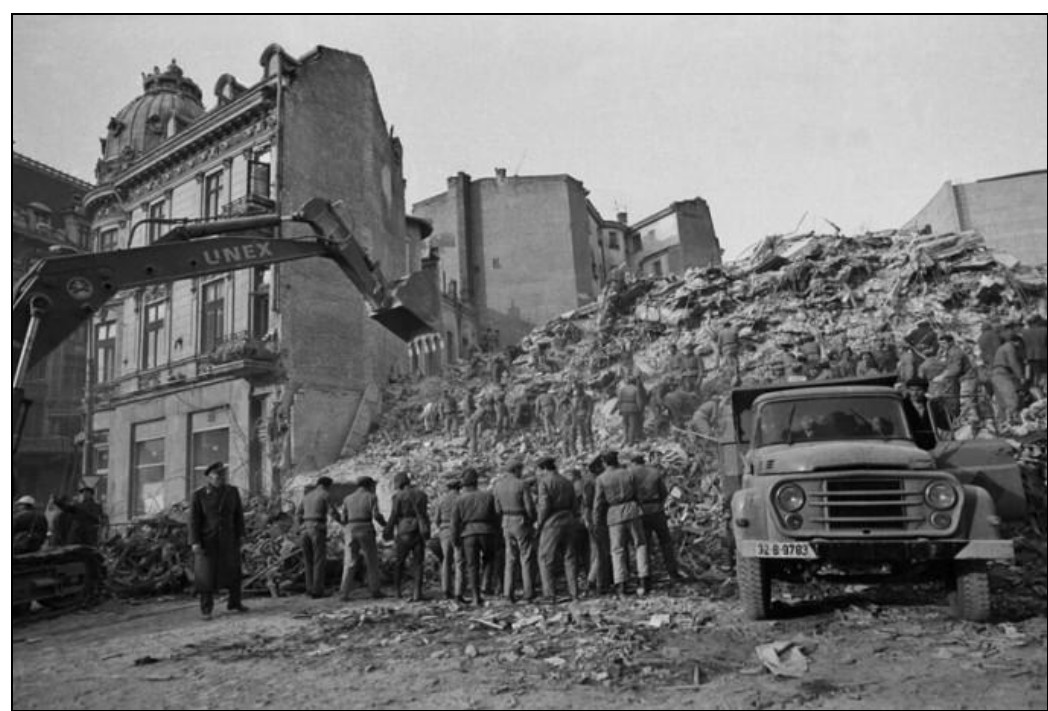

**Figure 2b. The collapse of the Continental block in Bucharest's historical centre in 1977 blocked the access streets, so clearance was delayed by more than 12 hours; the photograph shows efforts to identify victims and property.**

Source: Agerpres

Immediately after the earthquake of 4 March 1977, the former regime announced the start of a rehabilitation project for the highly-degraded buildings within the central area, a project which was abandoned in less than a year. Many buildings, after being braced in position for 6-7 months with wooden or metal poles (which were later withdrawn), were then only "cosmeticized" and reoccupied. These decisions set a precedent for irresponsible policy that, unless it is addressed and altered, could have disastrous long-term consequences.

Additionally, the growth of complacency over time has been a great enemy and a permanent state of vigilance is needed.  Finally, there is a need for considerable public investment in mitigation in the areas most vulnerable to earthquakes. The lack of wider public awareness  of the high seismic risk of these buildings (identified as a result of surveys conducted in and since the mid-90s) is evident in that the apartments in these 'cosmeticized' blocks are still the among the most expensive in the city due to their central location and  spaciousness.



Accordingly, the Romanian government has established a National Committee for Emergency Situations and a Department for Emergency Situations. The department coordinates the General Inspectorate of Emergency Situations. 41 local inspectorates, cover Bucharest city and the department of Ilfov. Bucharest city has three existing meeting points (two in Bucharest and other in Ciolpani village to the north) and there is a special strategy to bring in instantaneous support from 24 counties surrounding the city.

## 3 Data and Methods


    This assessment of seismic hazard and vulnerability includes quantitative and qualitative data analysis that incorporates physical, environmental, social and economic factors, potential impacts from existing risk maps and estimates of the population that would potentially be affected (Mândrescu, 1990; Armaş, 2012; Rufat, 2013; Pollino et al., 2012). The accessibility analysis takes into consideration the specificities of each urban district,

and especially the urban social context, too (Noto, 2017).

### 3.1 Data

    The authors have used several data sets (buildings classified by seismic risk and emergency categories, i.e. the presence of hospitals, and fire stations) in order to provide a realistic depiction of the impact that a potential

earthquake could have in the historical center of Bucharest. Only those fire stations and hospitals within the municipal limits of Bucharest are included. The main data sources were provided by public institutions. Every year the Municipality of Bucharest publishes a technical report classifying buildings with relation to four seismic risk criteria, and three emergency criteria to assess their level of public safety. Table 1 shows the distribution of Bucharest's buildings by risk and emergency categories. Special attention is paid to the historic

center of Bucharest which contains the largest concentration of buildings which represent a public safety risk.

    To represent the accessibility patterns prior and subsequent to an earthquake, it was necessary to digitize all elements of the transport infrastructure, construction, green spaces, alleys, sidewalks and property limits. Several different map sources were used to identify building locations including cadastral maps at scales of 1: 500, IGFCOT, 1: 2000 IGFCOT (1974-1975). Other map types and sources include old maps of Bucharest

produced by the Topographic Military Directorate and orthophotomaps (2014) taken from the National Agency for Cadastre and Real Estate Advertising. The authors have overlaid accessibility patterns on a numerical model of the land, given the absence of natural barriers, since the Bucharest municipality is located on a plain.



**Table 1. Building Condition Data for the Historic Centre of Bucharest**

(Number of Restored Buildings is at the historical centre level, and Fire stations and Hospitals at the municipality level). RI-RIV – Seismic Risk Categories; U1-U3 – Emergency categories.

|  | R I | R II | R III | R IV | U1 | U2 | U3 | Total |
|---|---|---|---|---|---|---|---|---|
| Buildings with seismic risk in Bucharest | 343 | 344 | 97 | 7 | 309 | 615 | 650 | 2,365 |
| Buildings with seismic risk in the historical centre of Bucharest | 65 | 1 | 2 | 1 | 82 | 9 | 5 | 165 |
| Buildings with seismic risk in the core of the historical centre of Bucharest | 50 |  | 1 | 1 | 41 | 2 | 2 | 97 |
| Number of buildings restored to an adequate seismic standard | 17 | | | | | | | |
| Fire Stations | 13 | | | | | | | |
| Hospitals | 32 | | | | | | | |

Source: *processed data using the List of Buildings with Seismic Risk, published by Bucharest Municipality*
170        (*http://amccrs.pmb.ro/docs/Lista_imobilelor_expertizate.pdf, accessed at October 15, 2017*)

**3.2 Methods**

An important methodological contribution on the capacity of a city to resume urban functions after a seismic event is the study by Goretti et al. (2014) on how the Crotone urban system could better respond to such
disasters. This study shows the importance of rapid accessibility to collapsed buildings and to injured people. Our study therefore emphasizes the importance of immediate accessibility for emergency intervention mechanisms, and the need to provide information to facilitate the proactive actions of decision-makers, who need clear and straightforward directions.

The main methodological steps in mapping accessibility in the central area of Bucharest were : a) setting up
a referenced database of all the buildings with seismic risk; b) transferring this information to a detailed map of the identified buildings; c) identifying indicators of building  density and age, and  traffic (including pedestrian)intensity ; d) showing the  locations of all hospitals, and fire stations; e) calculating  present-day





(before a possible earthquake) accessibility levels; f) identifying    specific locations of potential congestion resulting from the collapse of buildings included in the highest risk class; g) determining, by simulation, the

immediately inaccessible or poorly accessible areas   for the intervention crews in case of an earthquake occurrence, taking into account those buildings that might collapse if an earthquake occurs.

In the scientific literature, "access" is mainly measured as a physical distance or travelling time (Sotoudehnia & Comber, 2011). In this study, mapping the accessibility of the central area of Bucharest was completed using GIS techniques incorporating spatial analysis. The calculation of accessibility was initially based on the

geometric structure of the public transport network (busses, trams and underground services), and walking and cycling networks (Graeme & Aylward, 1999; Parker & Campbell, 1998; Naphtali, 2006; Svensson, 2010; Weiping & Chi, 2011; Sotoudehnia & Comber, 2011; ESPON TRACC Interim Report, 2013; ESPON GROSSE, 2013; Blandford et al., 2012; Coffee et al., 2012; Yiannakoulias et al., 2013; Vojnovic et al., 2014). The Kernel Density tool was used to calculate the density of point and line features in a neighborhood around

those networks. After modeling the road network using the ArcGIS Network Analyst extension, the authors used an assortment of analytical tools.  These included  the New Route tool  to check the road network; the New Closest option  to determine the closest emergency facility (hospital, fire station) to each point; and the New OD matrix function to determine optimal routes (depending on road distance and travel time) following the principle of the shortest possible route to establish links between each pair of points.

To highlight accessibility in the most comprehensive way, the street structure (which is very dense in the historic center where the streets are narrow) and road traffic density had to be taken into account. Accessibility was calculated as a function of the distances between different residential areas and hospitals and of the time necessary for these movements (using isochrones). Isochrones maps, showing travel times by public transport from the city centre, had been used to assist in urban transport planning in the 1950s (Kok 1951, Rowe 1953

quoted by O'Sullivan, et al., 2000). These isochrones were generated using geographic information systems (GIS).

Accessibility was also calculated to take into account the presence of specific service locations which could exacerbate the impact of potential disasters, such as gas stations and electric transformers (Rezaie & Panahi, 2015). In addition, the Kriging Kernel interpolation calculation and local polynomial interpolation were used.

For exact interpolation, the inverse distance weighted (IDW) method was used. These methods identified support elements for more proactive management that have the potential to bring about a decrease in both the material damage and the human casualties resulting from a strong earthquake. Using a database in a GIS environment enabled an assessment and estimation of the potential damage that could be caused by such an





event. At the same time, GIS is a valuable method of analysis for this purpose because the databases can be
regularly updated, allowing for ongoing mapping of the changing risk scenarios and the updating or reassessment of potential damage. The risk scenarios also provide useful identification of the vulnerable areas and population groups (Sinha et al., 2008).

The penultimate methodological step was to identify likely congestion locations. The initial simulation assumed that all the buildings categorized as possessing the highest degree of risk would collapse. For the core
of historical center, this permitted the identification of some important sites and street segments, which would be blocked in the case of a strong earthquake using the location of each highest risk building, their age, and number of floors and the local configuration of the street network.

Our intention is not to propose a precise correlation between the vulnerability of buildings (based on all their characteristics) and the intensity of the next earthquake. Rather, especially by taking into account that some of
buildings in this area have partially collapsed in the absence of a direct seismic cause; we contend that an earthquake of similar magnitude to the 1977 event would produce outcomes comparable to our simulation. .

From this information several maps were developed taking into account the region's particular seismogenic characteristics (Mäntyniemi et al., 2003). Two offer general images of accessibility at the city level closely correlated with the territorial distribution of fire stations and hospitals. Another identifies areas or street
segments potentially isolated by building collapses.

Our approach, focusing on the single issue of accessibility in a situation of crisis management, shows empirically how GIS technologies can be used to make recommendations to authorities to improve their preparedness levels and response speeds in post-earthquake interventions. Within this study, GIS is used solely as a tool to identify accessibility as a starting point for disaster management (Nushi & van Loenen, 2013). These
GIS solutions are demonstrably important applications in relation to the first two phases (risk mitigation and disaster preparedness) of Alexander's (2002) four-phase sequence of emergency management activities,

## 4 Results and Discussion

It is necessary to simulate emergency interventions prior to the occurrence of catastrophic events because, in the local situation, the inherited intra-urban structure, with a narrow winding street pattern  dating back to
medieval times, the poor structural condition of many of the buildings, and  limited access to important points from  the emergency response activity locations are all of critical importance.

In such a context, accessibility to specific disaster sites is critical and this requires that urban areas of this nature be treated with special attention. The biggest challenge may be caused by traffic congestion compounded


by debris, which can isolate critical areas, making rapid intervention to put down fires and save human lives
impossible.

The official identification of buildings with a high seismic risk combined with precise mapping of their
location (Fig. 3) can be related to the density of road traffic in the historic areas (Fig. 4).  If traffic is very high
on the main access streets this could inhibit rapid intervention, especially in a situation general panic such as
that generated by a potential earthquake. It would also be difficult to use narrow streets, where the pedestrian
traffic and partially collapsed buildings could block the access of emergency service vehicles. In this context,
there is a need for proactive measures, to mitigate the risk of late arrival of assistance at the affected buildings.

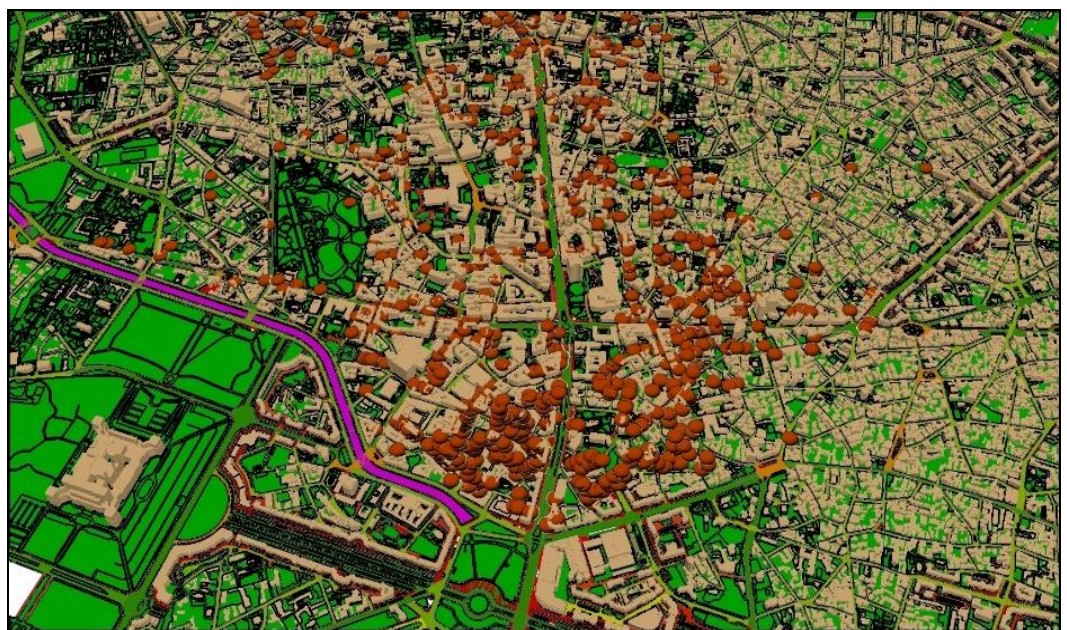

**Figure 3. Location  of  buildings with a high earthquake risk**

The most important area of the historical center is the one delimited by Armenească, Moşilor, and Călăraşi
streets, Splaiul Independenţei, Calea Victoriei, and Carol and Regina Maria boulevards. Within this area, the
building density exceeds 2.5 units / hectare and, in some places, even 10 units / hectare.  In the areas of the
highest density, most of the buildings have two or three floors, and, because of their uncertain legal status after
1990, many   exhibit an advanced and increasing degree of dilapidation. Restoration and reinforcement of these
buildings by both public authorities and private entrepreneurs is only proceeding at a maximum rate of 2
buildings per year.



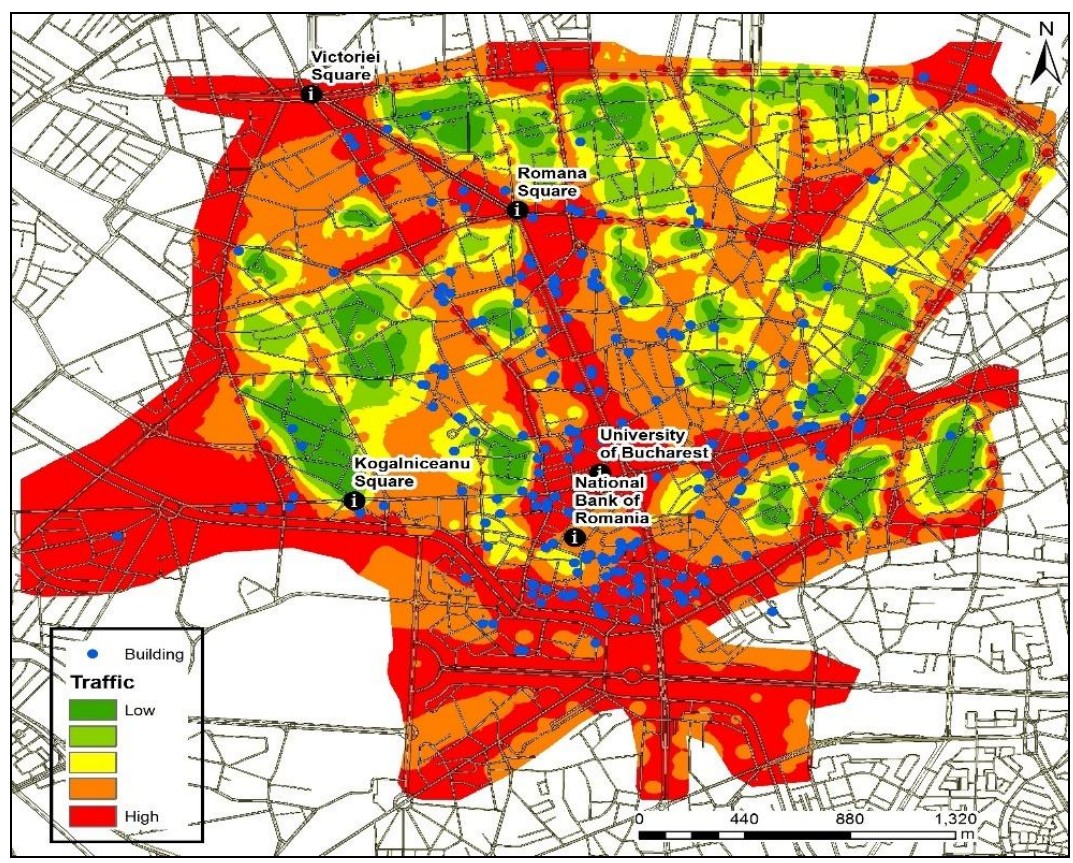

**Figure 4. Road traffic density and the location of buildings with a high seismic risk**


The number of buildings with the highest seismic risk (computed with Kernel Density tool) shows a very high concentration in the historical center of Bucharest (Fig. 5). Looking at a map of seismicity at the level of Bucharest, it becomes obvious that the inherent risks from earthquake damage are greatest in central Bucharest, including the historical center (Rufat, 2011). Even though most of the buildings located in the historical center

date from the early 20th century, they were built on the foundations of 19th century structures (Armaş, 2008).





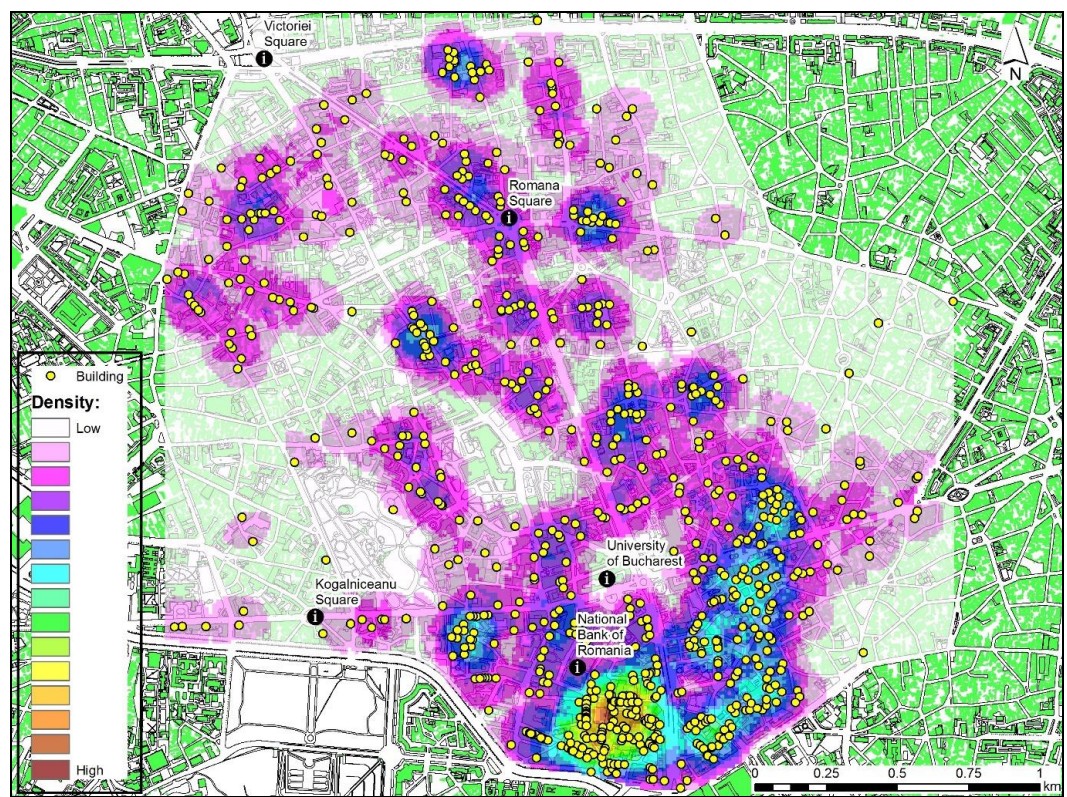

**Figure 5. Density of buildings with a major seismic risk**

To highlight the anticipated degree of access for fire protection and ambulance services in the central area,
accessibility levels prior to an earthquake were calculated and later compared to a post-earthquake scenario.
Taking into account the location of the fire stations and hospitals, and the street tram network, the access routes,
into and within the study area were evaluated using as a Network Analysis tool. Thus we identified the shortest
routes from the closest emergency facilities (fire stations and hospitals) to all locations in the study area, using
the Bellman-Kalaba algorithm. These minimal road applications were applied for various types of emergency
service, such as transport, ambulance, fire, police etc.

It was noted that both firefighting and ambulance service accessibility were high or very high for most parts
of the capital city, including the downtown area, which is especially well served by firefighter and ambulance
services. There are 13 large fire stations in Bucharest. However, the lowest levels of potential accessibility by
fire services to individual houses in Bucharest city, occurred in the historic center area, mainly due to the
configuration of the street pattern (Fig. 6). The lowest values were registered in an area between Calea Victoriei,



Doamnei Street, Brătianu and Splaiul Independenței, in the core of the historic center. Low values occur to the East of Bratianu Boulevard, even though some important access axes (Armenească, Calea Moșilor, Hristo Botev, Negustori) are located nearby. Overall, if fires broke out at several different points in the historic center core during a seismic event, this would present huge problems.

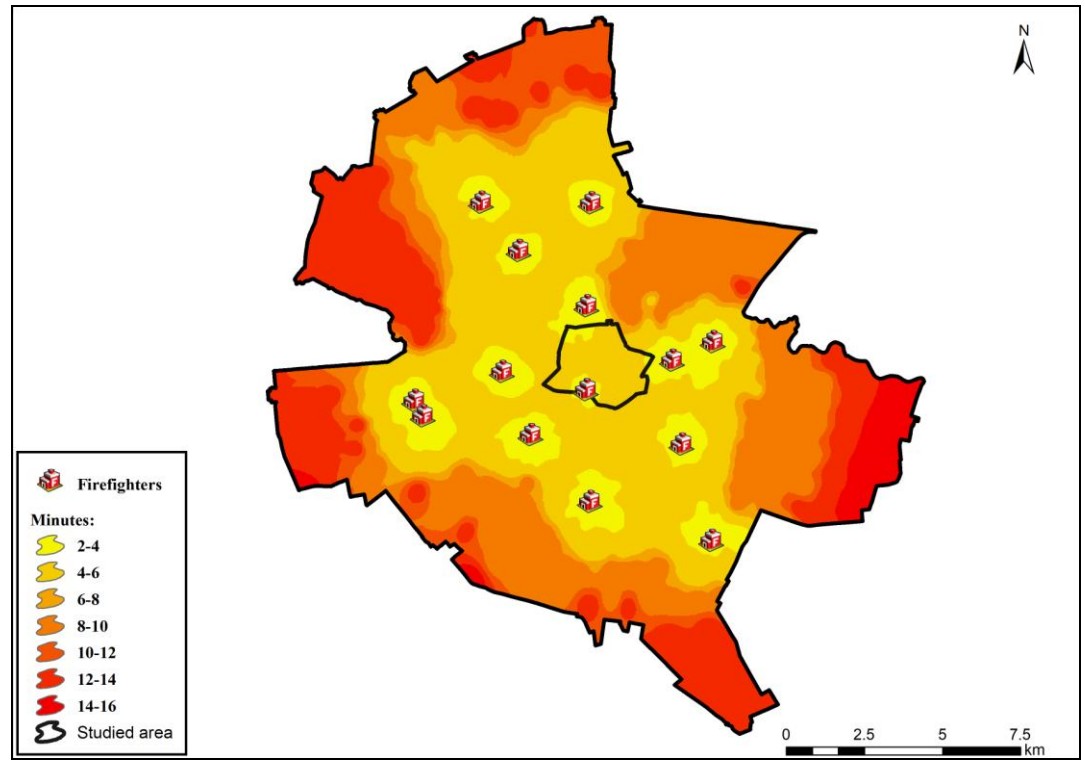

**Figure 6.  Fire fighting accessibility levels within Bucharest city**

The map of ambulance accessibility (Figure 7) presents a very similar picture.  However, access is better in the northern part of historic center due to the location of Colțea Hospital. Figure 7 shows two areas where it would be difficult for ambulances to arrive in a timely manner, one in the core of historical center and the other in the surrounds of Mihai Vodă Street, which would be accessed by ambulances from University Hospital.



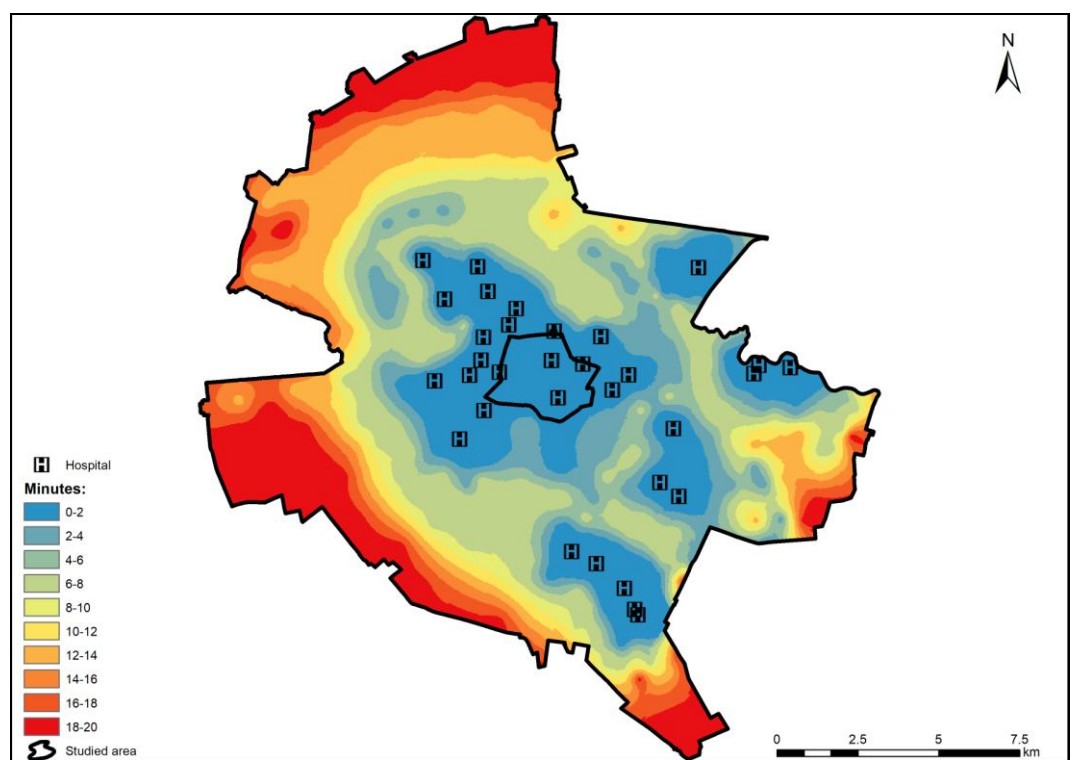

**Figure 7. Ambulance accessibility levels within Bucharest city**

Should an earthquake occur, an important consideration is the challenge presented by building collapses which obstruct road access. Identifying individual buildings with the highest levels of seismic risk highlighted the possibility of concentrated building collapses in certain locations within the historic center. In these the locations, some buildings would become isolated and rapid intervention by fire or ambulance services would be impossible.

This general analysis indicates that the central area seems to be favored due to the possibility of intervention from several emergency service points into this part of the city. However, and in spite of this, pedestrian and vehicular congestion is highly likely to inhibit rapid access by firefighters and ambulances in several areas within the central district. Also, several locations in the downtown area, which previously appeared to have high emergency accessibility levels, were shown to possess high probabilities of multiple building collapses. These

events could well obstruct access by emergency vehicles, despite the high levels of accessibility that were identified initially.



Should an earthquake with a magnitude of over 7 degrees on the Richter scale occur, fires would present a major associated risk. The majority of the city center buildings is of timber construction or possesses many timber components (some buildings from "Şelari" "Crama Domnească"; "Covaci" and "Smârdan" streets, for
example). These buildings characteristically house restaurants, cafes or pubs, which contain huge quantities of furniture, a further important source of fire. If emergency action does not occur promptly, in such locations, the probability of numerous fatalities is high. In addition, water supply and sewerage systems may be damaged, resulting in basement and ground floor flooding. It would therefore be advisable to provide supplementary emergency response materials at a large number of locations within this district. This would allow access to
such equipment at the local scale as an alternative to the provision of emergency materials and services from elsewhere which may be unobtainable in the event of an emergency.

Assuming that, in the event of a large-scale disaster, certain clusters of buildings may become isolated and inaccessible to emergency services it is therefore recommended that smaller scale aid stations be established within these districts. These smaller scale aid stations could then provide critical assistance in areas isolated by
building collapses.

An in-depth and more detailed study of a portion of the general study area close to several buildings of national importance (the Parliament building, the headquarters of several ministries and other public institutions) since the location of emergency services may be oriented to the protection of these public institutions, rather than to the provision of services to areas with high densities of buildings with high seismic
risk.

The map which shows both the distribution of the highest seismic risk buildings and the location of the nearest fire stations (Fig. 8), illustrates the need for greater proximity (and hence access) of fire stations to the two areas of maximum density of highest risk buildings: one in the Lipscani area and the other in the Bărăţiei area. The Western area (Griviţa - Gara de Nord) could be placed under the authority of the two existing fire
stations. These areas of high vulnerability should be connected to a permanent emergency water supply (since the normal water sources would be disrupted by an earthquake). They should also possess a minimum, yet sufficient, level of equipment for a local first response.





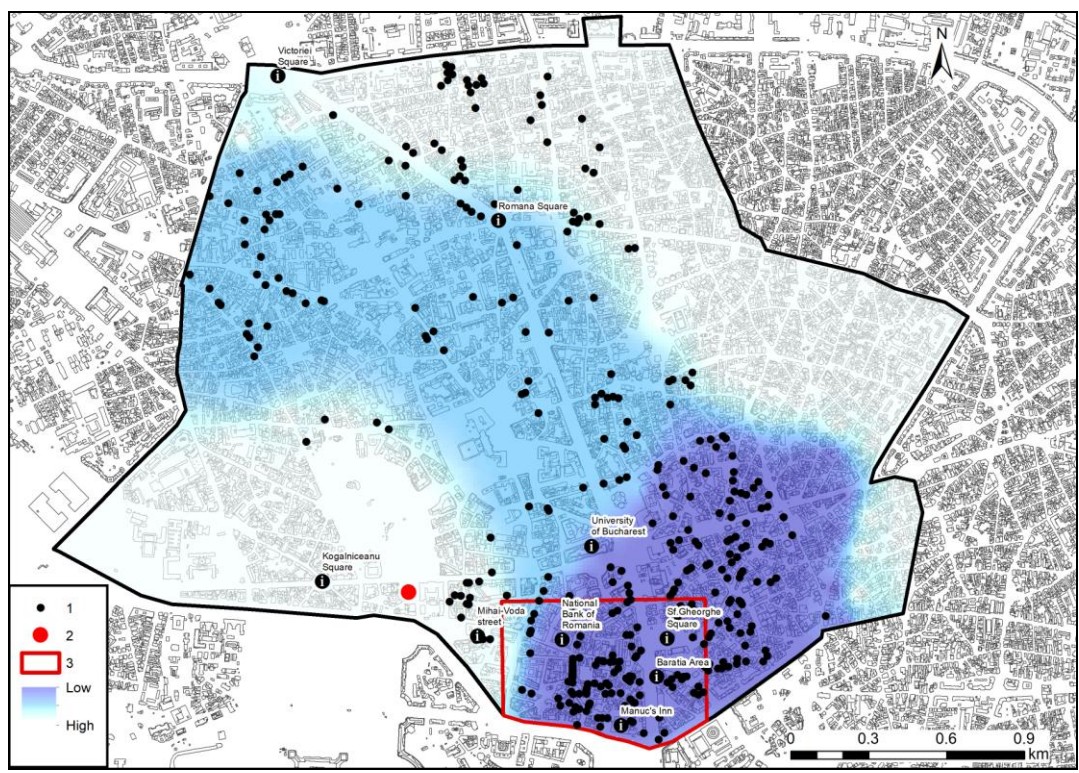

**Figure 8. The accessibility of buildings with high seismic risk to fire fighting services after a potential**
**earthquake occurrence.** 1. Building; 2. Fire station; 3. Core of historic centre


The location of hospitals again appears to be favorable at first glance (Fig. 9), but their capacity should be assessed against the probable number of casualties, which could reach as high as 11,000. The earthquake of 1940 registered 1,271 and the 1977 earthquake 11,321 injured persons (Pavel & Văcăreanu, 2015). The location

of the Hospital Colțea suggests that the majority of injured persons would go there for immediate treatment. However, this hospital only possesses a small surgical unit (with three operating theatres) and it would be unable to offer emergency medical assistance to a large number of persons over a short period.



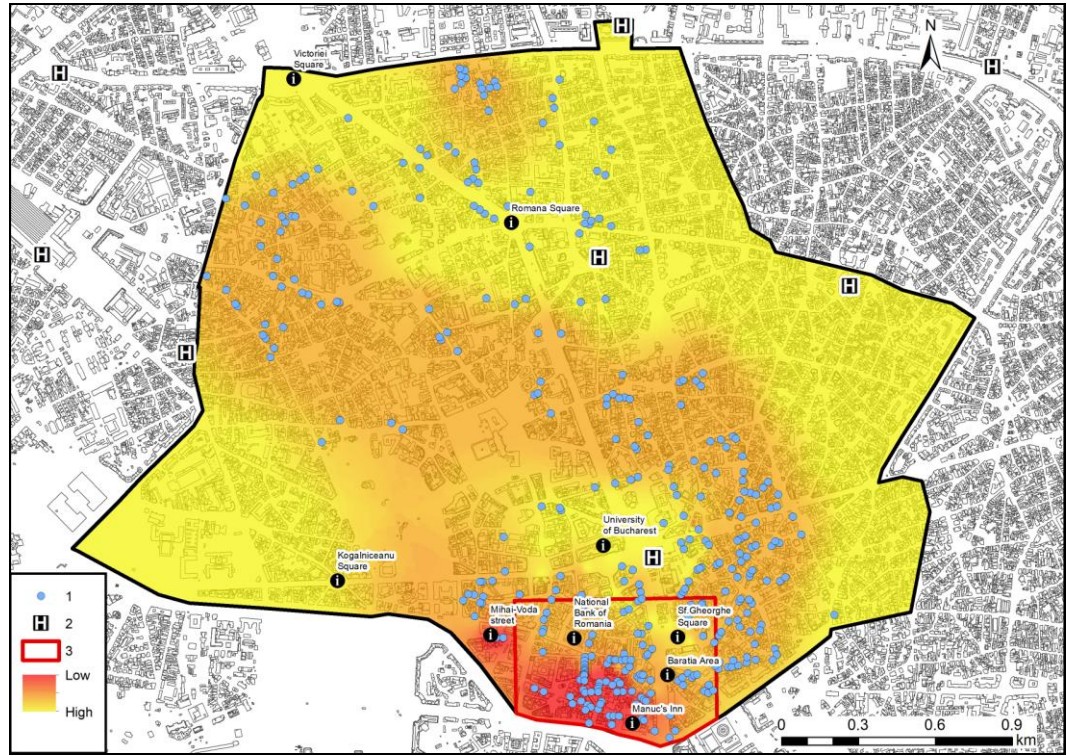

**Figure 9. The territorial distribution of hospitals in the central area and their relationship to buildings with a high seismic risk.** 1. Building; 2. Hospital; 3. Core of the historic centre

To increase the efficiency of emergency response, the location and number of potential casualties must be more precisely determined. Consider needs to be given to the availability of specific medical services at
individual hospitals and other medical facilities. The provision of surgical wards, imaging laboratories, and orthopedic facilities is more uneven than the provision of hospitals more generally across the city. Depending on the territorial distribution of these specific hospital services, the buildings with highest seismic risk should be assigned to specific emergency hospital services so that accessibility levels can be maximized. Obviously, this implies the designation of dynamic territorial structures, which, depending on the gravity of the reported seismic
events and their human consequences, would include access to other hospitals at greater distances from the central area (Toma-Dănilă, 2013).

In the event of a powerful earthquake, a partial or even total breakdown of communication systems is likely. This eventuality would cause many people to make direct contact with friends and relatives by moving around




the city by car. Rapid intervention by the traffic police will be vital to minimize congestion in those areas of the
city where the need for emergency intervention is greatest.

    The unpredictable nature of this phenomenon may well lead to traffic bottlenecks at unanticipated locations along the transportation network, which in turn would further complicate rescue, relief, and evacuation efforts. In these circumstances, communication systems between those who would be mapping the collapsed or damaged buildings and those who would be ensuring the traffic flow need to function as smoothly as possible in
order to allow the wounded to be transported to hospitals and the fire engines to move towards critical spots in the city. Where the simultaneous collapse of buildings, especially in the medieval area of the city, made rapid intervention impossible, lifesaving equipment, individually transported by specially trained persons, would be needed provide immediate assistance.

    Our study has sought to demonstrate what could happen in the core of historic center (Fig. 10), taking into
consideration the likely collapse of buildings classified as Risk 1 (R1). Any future earthquake of more than 7.2 on the Richter scale (the level of the strongest recent earthquake on March 4[th], 1977) would pose an amplified danger of the collapse of buildings in critical locations. We define critical locations, as those where building collapse, could block access to specific areas or street segments.

    These potential blockages are most likely to occur in 5 areas, identified in Fig 10 as A, B, C, D. E. *A*
(Blănari area) is small and is delimited by building no.2 (built in 1865, 3 floors), and the buildings from no. 9 (1880, 5 floors) to no.14 (1935, 6 floors). *B* (Lipscani-Gabroveni area) is the largest site, and contains a group of 15 vulnerable buildings. Possible street blockages could be produced by the collapse of buildings on Lipscani Street, such as no.26 (1864, 5 floors) or no.29 (1934, 9 floors), and no.76 (1906, 4 floors), and on Gabroveni Street no.2 (1940, 9 floors) and no.12 (1924, 6 floors). Two other areas, *C* and *D,* are located on Franceză Street
between buildings no.6 (1869, 5 floors) and no. 22 (1900, 6 floors), no. 30 (1870, 5 floors) and no. 42 (1870, 5 floors). On the West side of the main boulevard Brătianu, *E* area (Bărăției) contains the buildings no. 8 – Baia de Fier Street (1930, 5 floors) and no. 37 (1870, 3floors) and no.50 (1824, 4 floors) Bărăției street.

    A final consideration is that, for much of the day, the core of the historic center normally contains between 1,000 and 5.000 visitors in addition to the area's residents and workers. This only adds to the need to devise
proactive earthquake intervention and mitigation strategies.





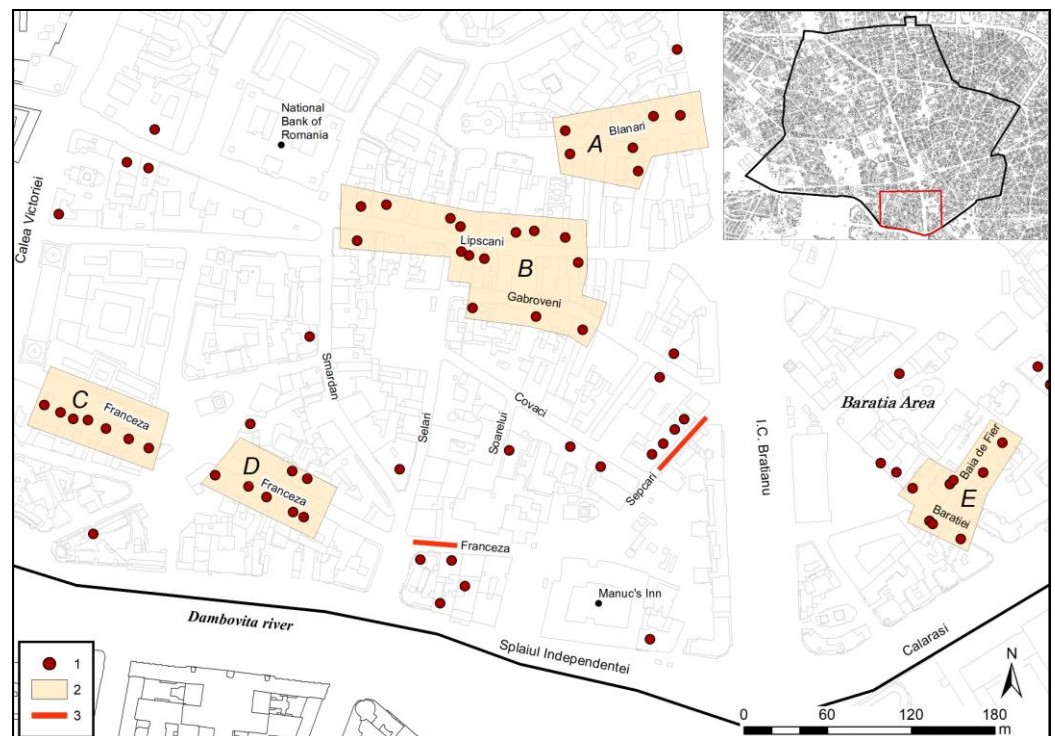

**Fig.10. The anticipated spatial effects of building collapse in a similar earthquake to that of March, 1977.**
1. Building in the first category of Risk (R1); 2. Isolated areas resulting from the hypothesis that all buildings belonging to the R1 class would collapse; 3. Blocked street segments.

We therefore suggest the following proactive measures to mitigate the risks associated with a seismic disaster in Bucharest city, and especially within the core of the city's historic center:

- the constitution of a technical team of decision-makers to identify optimal response strategies for a future earthquake. The main task of this team would be to identify the critical points and areas for emergency intervention in the most congested areas (Tuns et al., 2013).

- the prioritization of building consolidation, correlated with the buildings' locations and their potential to block street segments and critical access routes in the event of their collapse ;

- the re-evaluation of the number and locations of fire stations

- the development of a system of emergency medical aid posts within the historic center, taking into account both the area's access problems and the fluctuating population of the area resulting from its entertainment role.



- reorganization of the "Colţea" Hospital, including the expansion of its infrastructure (especially the surgical section, and the number of operating theatres). This hospital should become the most important point of emergency intervention in the historic centre of Bucharest in the event of a strong earthquake.

**5 Conclusions**

This study demonstrates that GIS can be used effectively as an analytical and decision making tool in planning for hazard mitigation. GIS, properly employed, can provide information concerning emergency response accessibility in areas where physical structures are degraded and pose a higher risk of collapse. Such knowledge is critical in anticipating the impact of a disaster. Injury, loss of life, and damage to property can be minimized through more effective and rapid emergency response.

In Europe, Bucharest ranks second, after Istanbul, in its exposure and risk related to earthquakes. It is not enough to be familiar with the distribution of the high seismic risk buildings. Emergency intervention is also vital to minimize the consequences of such an event. In order to save lives, knowledge of accessibility levels and related rapid intervention potential is essential in the event of an earthquake. However, the current status and priorities of natural hazard and emergency response planning in the city of Bucharest (and at the national
scale), are such that they are unlikely to mitigate the effects of a potential disaster to a sufficient extent. Across several measures – the training of specialists, public awareness and education, infrastructural improvements, and building improvements – current efforts are inadequate.

    The passivity of urban decision-makers in relation to the very large number of buildings in the highest risk class perhaps the most surprising element here. These buildings are concentrated in the most populated and
attractive areas of the city in terms of leisure and entertainment. Even if Bucharest's inhabitants are partially aware of this risk, the vast majority of tourists are unlikely to realize what could happen should an earthquake occur.

    The most recent major disaster event that took place in Bucharest, the fire in the „Collectiv" Club on the night of October 30, 2015, which led to the deaths of 63 people and serious injury to around 150, brought the major
risk that an earthquake can pose to the attention of the local and national authorities. In response, a ban was imposed on all shops, restaurants and clubs operating in buildings with high seismic risk, but there are not enough resources to refit the high risk buildings that continue to be inhabited. This event reminded the population and the authorities that an earthquake event or disaster of similar scale will occur again at some point



and that it is necessary to have a clearly defined policy that relies upon concrete measures to reduce the human

and material losses.

Our study reveals both the importance of accessibility to buildings for emergency intervention, and the shortcomings in the current provision of major emergency response services. Our methodology, using simple tools, offers analysts and decision-makers a credible means of developing a proactive vision of and a management strategy for emergency response in congested, historic areas. GIS is a commonly used tool for

analyses of this type and its results, since they can be expressed cartographically, can be more widely understood than is often the case when other statistical and computational techniques are employed.

***Author contributions.*** CM an II designed the study. GM, CM and II established and set-up the maps. II, CM, RJ, GM and GP analysed and interpreted the results. II and CM wrote the paper with substantial input from all

co-authors. RJ revised the English.

***Competing interests.*** The authors declare that they have no conflict of interest.

***Acknowledgements.*** This work has been partially supported by the Project UB 2008.

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
