# Peer review of "Mapping Accessibility for Earthquake Hazard Response in the Historic Urban Centre of Bucharest"

_Natural Hazards and Earth System Sciences, 2018_

## Referee Comment (RC1) · Anonymous Referee #1 · 10 Mar 2018

Abstract

The abstract summarises well the article. The aim is clear from the title and the abstract.

Background

The paper has a clear research question on accessibility in case of emergency after earthquake in a mixed street pattern city centre. To previous studies, either known by the author or referenced the paper adds a new case study in a city less covered by literature. Compared to the previous version, the references improved. However, technical correction to Crowley, H., Colombi, M., Pinho, R., Meroni, F., and Cassera,

A.: Application of a prioritisation scheme for seismic intervention in school buildings in Italy, in: 14th World Conf. Earthq. Eng., Beijing, China. Although the WCEE papers are archived in the web, there is a better referenceable paper by the authors in Earthquake Spectra (Damian N. Grant, Julian J. Bommer, Rui Pinho, G. Michele Calvi, Agostino Goretti, and Fabrizio Meroni (2007) A Prioritization Scheme for Seismic Intervention in School Buildings in Italy. Earthquake Spectra: May 2007, Vol. 23, No. 2, pp. 291-314.)

Methodology

The paper develops a methodology to deal with GIS tools for the case study of Bucharest. Bucharest is a large city in Europe (2 Mio. inhabitants) posing sesismic risk. At the same time, since more than 40 years passed since the last damaging earthquake, awareness to seismic risk decreased. Many high seismic risk buildings, though identified, are not being retrofitted, which would lead to loss in an earthquake. This is what the paper focuses on: the emergency planning in case that an earthquake would occur. In this sense it is helpful to civil protection. The paper improved compared to the last version in the explanation of fire risk. The density of high seismic risk buildings in the historic centre compared to the Magheru boulevard (where high density reinforced concrete buildings are posed to hazard) has been properly highlighted with corresponding maps.

Data and Results

The study matches the results as presented briefly in the abstract. The paper properly underlines the study results with a table and graphs. Fig. 10 presents a relevant result with blocked street from the collapse of high density vulnerable buildings in the historic centre. Although historic earthquakes lay so long back, the paper properly includes recent events which raised awareness on hazards, such as the Colectiv fire, hence the inclusion of fire hazard is important.

As results, there are useful recommendations for decision makers, which build properly on the civil protection studies from Italy which were mentioned, and

also on the reference to Schweier and Markus who collaborated on emergency issues with the Romanian civil protection. This is a reason I propose it for highlight. The Frank Fiedrich article I suggested within the same collaboration is for example http://ieeexplore.ieee.org/abstract/document/4117644/?reload=true or https://www.informs-sim.org/wsc06papers/059.pdf which included simulation of post-earthquake fire for Magheru boulevard in Bucharest. More recent writings adressing urban infrastructure such as roads by the author are Urban Disaster Resilience and Security. Addressing Risks in Societies. Editors Alexander Fekete Frank Fiedrich (Springer) and Einführung in den Bevölkerungsschutz. Autoren: Fiedrich, Frank, Kudlacek, Dominic (Springer), the second might be more difficult to understand as it is in German but it means "Introduction to population protection".

The maps have a better visibility in 2D as in 3D in the initial article after the consideration of the reviewer.

In addition to the letter of the authors we note that also the main author, although initially included, changed and was responsible to set up the maps.

In summary the article improved and took in consideration the observations from the previous revision (there was answer to the reviewer comments, also considering some of the comments for a future study) and I recommend it for publication. I think that it is especially relevant for the civil protection involvment in emergency planning, which is one of the four planning models in disaster management, if preparedness planning was incompletely approached as it is the case in Bucharest, and mitigation and resilience planning are not yet mature. The European Commission has special funding programmes for civil protection involvment in disaster management. I recommend only technical corrections regarding the references.

---

## Author Comment (AC1) · 14 Mar 2018

We thank the Anonymous Reviewer for the careful attention given to our paper, and, especially, for their positive comments on the improvements made to our previous version. We are encouraged by the suggestion that our methods could also be applicable in other contexts and areas. Certainly, we see potential in extending this type of investigation to those densely built up residential districts of Bucharest where only narrow spaces have been left between the high blocks of workers' flats.

We are also appreciative of the suggestions of additional and up to date references. These will be incorporated in our revisions to the paper and will be valuable for any

extension of our work to further areas of the city. Finally, we note the reviewer's corrections to our list of references and we will rectify these irregularities as part of our revisions to the paper.

---

## Short Comment (SC1) · 30 Mar 2018

A. General considerations

The present-day literature regarding to civil protection in the case of such events, and, especially, in Romania, is not rich. I consider that the paper is welcome, because could push the scientists and decision-makers to work better for solving the urgent problems of communities. At the same time, comparatively with the previous version, this one is well written, in good English. The authors have avoided some confusions, have changed some maps (replacing the 3D with 2D ones), and have improved the analytical study.

[Figure]

Abstract In some words, the authors succeeded to make a synthesis on their research, defining the goal of study, methods and results.

Introduction The authors achieve acceptable literature documentation, focused on the proactive management and the accessibility for emergency situations. Preparedness the people for a future strong earthquake asks a rapid accessibility to the collapsed buildings for mitigate the human and materials losses. Using GIS technologies they individualize the potential affected areas and the problems for accessibility.

Methodology A detailed description of the methodology is offer, starting with the careful field analysis of the studied area, using a selective analysis of the statistical information, and finalizing with the simulation maps for a rapid accessibility of ambulances and firefighters interventions.

Results Using vectorial and spatial data included in the GIS soft, the authors depicted critical areas and point in the case of a strong earthquake. In their concrete analysis, they used the buildings, technically investigated and included in the seismic risk I class. Their territorial distribution in the historical center of Bucharest and the location of hospitals and firefighting stations were the fundamental elements for accessibility maps at different scales. These maps could fundament the decision-makers actions for the building of appropriate emergency plans.

B. Specific remarks:

1) Starting from the fact that this is a case study made on the core of historical center of Bucharest, and, taking into account only the seismic Risk I buildings, we recommend to authors, may-be in the next research, to extend the analysis at the Bucharest level, and to other seismic risk classes of the buildings. There are many such buildings in other Bucharest areas, which need to be analysed from the point of view of accessibility. Such type of analysis is necessary to be known by the public administration offering more cartographical materials, which could be used for their future actions, and for good information of the citizens. 2) Suggestions and corrections a) Suggestions: - the

avoiding of the comment to the Fig. 2b, respectively: "the photograph shows efforts to identify victims and property"; - the streets canevas on the Fig. 10, is not clear. Please, make clearer, for a better orientation of the readers! b) Corrections: - For uniformity of English, please replace in the entire manuscript all centre(s) with center(s), or inversely. See the followings lines: 120,165,166,168 (inside the table), 204, 340, and 412.

Appreciating this research, I consider that this version is original one, and can be published in the present-day form, with the mentioned suggestions and corrections.
* * *

---

## Author Comment (AC2) · 5 Apr 2018

We note and appreciate that this comment acknowledges both the strengths of our paper and the improvements that we have now made to the original version.

We also note and appreciate the suggestions for further improvements. We will alter Figure 2 in line with these suggestions and improve the quality of Figure 10. We will also ensure that UK English style and spelling is used throughout in the revised version of our paper.

2018-41, 2018.

---

## Referee Comment (RC2) · Anonymous Referee #1 · 5 May 2018

Dear Cristina Merciu,

thank you for addressing my suggestions and improving the reference list. Indeed the book in German by Frank Fiedrich is not yet available, as I also noted later on, the author is still working on it as a later conversation revealed. I apologise for not having checked it properly. The paper now fully corresponds to my suggestions and looking forward to see it published.

---

## Short Comment (SC2) · 5 May 2018

1. Referee suggestion 1:

Compared to the previous version, the references improved. However, technical correction to Crowley, H., Colombi, M., Pinho, R., Meroni, F., and Cassera, A.: Application of a prioritisation scheme for seismic intervention in school buildings in Italy, in: 14th World Conf. Earthq. Eng., Beijing, China. Although the WCEE papers are archived in the web, there is a better referenceable paper by the authors in Earthquake Spectra (Damian N. Grant, Julian J. Bommer, Rui Pinho, G. Michele Calvi, Agostino Goretti, and Fabrizio Meroni (2007) A Prioritization Scheme for Seismic Intervention in School

Buildings in Italy. Earthquake Spectra: May 2007, Vol. 23, No. 2, pp. 291-314.)

Authors' Answer: We agree with your recommendation. The article published in Earthquake Spectra is a more accessible reference, and the content is the same.

We have replaced Crowley et al., 2008, with Grant et al., 2007 in the main text, at the line 53.

The same change has been made in the REFERENCE list as follows:

Crowley, H., Colombi, M., Pinho, R., Meroni, F., and Cassera, A.: Application of a prioritisation scheme for seismic intervention in school buildings in Italy, in: 14th World Conf. Earthq. Eng., Beijing, China. ftp://ftp.ecn.purdue.edu/spujol/Andres/files/09-01-0097.PDF, Oct. 12-17, 2008,

replaced by

Grant, D.N., Bommer, J.J., Pinho, R., Michele Calvi, G., Goretti, A., and Meroni, F. (2007) A Prioritization Scheme for Seismic Intervention in School Buildings in Italy. Earthq. Spectra, 23, 291-314, 2007. https://doi.org/10.1193/1.2722784

2. Referee suggestion 2:

The Frank Fiedrich article I suggested within the same collaboration is for example http://ieeexplore.ieee.org/abstract/document/4117644/?reload=true or https://www.informs-sim.org/wsc06papers/059.pdf which included simulation of post-earthquake fire for Magheru boulevard in Bucharest.

Authors' Answer: Thank you for this recommendation! Inclusion of the paper by Frank in our references provided us with the possibility of adding some further relevant information.

The authors have added new material to the text as specified below:

a) At line 60, we have inserted: As Fiedrich (2007) suggests the responses made to

[Figure]

a disaster during the first three days are fundamental. After that, the main goals are invariably rescuing trapped victims, and treatming of the injured persons, though fire fighting may also continue in some cases.

b) At line 289, we have inserted: There are some studies of fire fighting simulations outside the historical center of Bucharest, in the Magheru Blvd (for example), which reflect the importance ascribed to this phenomenon during an earthquake event (Fiedrich, 2007).

The following REFERENCE has been added:

Fiedrich, F.: An HLA-Based Multiagent System for Optimized Resource Allocation After Strong Earthquakes, Simulation Conference, 3-6 Dec.,WSC 06, Proc. Winter, Monterrey, CA, USA, 2006, added to IEEE Xplore: 05 March 2007, DOI: 10.1109/WSC.2006.323120

3. Referee suggestion 3:

More recent writings adressing urban infrastructure such as roads by the author is Urban Disaster Resilience and Security. Addressing Risks in Societies. Editors Alexander Fekete Frank Fiedrich (Springer).

Authors' Answer: Your suggestion has added to our knowledge of this topic, since this publication provides additional confirmation of the importance of accessibility in the event of an earthquake. We have therefore made the following additions to the text:

At line 236, we have inserted: a) In recent years, scientific approaches to risk reduction in natural disasters, such as earthquakes have used resilience as an important concept which could provide new theoretical insights and practical measures for the enhancement of civil protection (Fekete and Fiedrich, 2018). Scientists, working with decision makers and communities, can use this concept profitably (Anhorn, 2018). Furthermore, Fekete and Friedrich's work has relevance to a range of issues related to accessibility levels in disaster affected affected areas.

The following REFERENCES have been added:

Anhorn, J.: Nepal and the "Urban Resilience Utopia", in Editors: Fekete A., Fiedrich F (eds.), 2018, Urban Disaster Resilience and Security, pp. 13-26. The Urban Book Series, Springer, 2018. https://doi.org/10.1007/978-3-319-68606-6.

Fekete, A. and Fiedrich, F.: Introduction to "Urban Disaster Resilience and Security – Adressing Risks in Societies", in Editors: Fekete A., Fiedrich F (eds.), 2018, Urban Disaster Resilience and Security, pp.1-12. The Urban Book Series, Springer, 2018. https://doi.org/10.1007/978-3-319-68606-6.

At line 32, we have inserteded: b) In any disaster situation, one of the most important factors across all the disaster phases is public-private emergency cooperation. By developing a model to harmonise this joint cooperation, Wiens et al. (2018) identify efficient ways to improve the logistics operations during crisis management.

to the following REFERENCE has been added:

Wiens, M., Schatter, F., Zobel C.W. and Schultmann, F. in Editors: Fekete A., Fiedrich F (eds), 2018, Urban Disaster Resilience and Security, pp.145-168. The Urban Book Series, Springer, 2018. https://doi.org/10.1007/978-3-319-68606-6.

4. Referee suggestion 4: Referee 1 also recommends another recent publication: . . . . . . . . .. Einführung in den Bevölkerungsschutz. Autoren: Fiedrich, Frank, Kudlacek, Dominic (Springer)".

Authors' Answer:

This potentialy interesting book is not yet available. However, we look forward to using its ideas in our future endeavours. The authors again thank Referee 1 for his/her close reading of our work and for his/her useful suggestions.

Please also note the supplement to this comment:
https://www.nat-hazards-earth-syst-sci-discuss.net/nhess-2018-41/nhess-2018-41-

SC2-supplement.pdf

**Supplement:**

[revised manuscript text omitted]

---

## Short Comment (SC3) · 5 May 2018

A. Suggestions:

1. Suggestion 1:

The avoiding of the comment to the Fig. 2b, respectively: "the photograph shows efforts to identify victims and property"

Authors'Answer:

We agree.

[Figure]

The caption for fig. 2B (page 6, lines 122-123) was therefore altered from:

Figure 2b.The collapse of the Continental block in Bucharest's historical centre in 1977 blocked the access streets, so clearance was delayed by more than 12 hours; the photograph shows efforts to identify victims and property.

To:

Figure 2b.The collapse of the Continental block in Bucharest's historical centre in 1977 blocked the access streets, so clearance was delayed by more than 12 hours.

2. Suggestion 2:

The streets canevas on the Fig. 10 is not clear. Please, make clearer, for a better orientation of the readers!

Authors'Answer:

We have improved the Fig.10 (page 20), making the street network more visible.

B. Corrections:

S. Boengiu' comments:

For uniformity of English, please replace in the entire manuscript all centre(s) with center(s), or inversely. See the followings lines: 120,165,166,168 (inside the table), 204, 340, 352 and 412.

Authors'Answer:

The spelling of "centre" has been changed to "center" as requested in the specified locations.

Finally, the authors again offer their thanks for the consideration given to our paper, and for the useful suggestions and corrections.

Please also note the supplement to this comment:

https://www.nat-hazards-earth-syst-sci-discuss.net/nhess-2018-41/nhess-2018-41-SC3-supplement.pdf

**Supplement:**

[revised manuscript text omitted]

---

## Referee Comment (RC3) · Anonymous Referee #2 · 16 May 2018

The authors present a study on the post-disaster accessibility of the historical centre of Bucharest under the assumption that all buildings with a certain risk classification undertaken by local authorities would collapse during an earthquake in the city. Accessibility was computed by means of GIS using the geometric structure of the transport network and considering possible network interruptions as the result of a seismic event. Accessibility was calculated as a function of the distances of different residential building areas from hospitals and fire stations. As such, the topic is of relevance for the target journal.

**1 Introduction**

Compared to the previous version of the article, the introduction has improved significantly. The introduction of the article outlines the connection between urban structures/development and seismic hazards emphasising the value of seismic risk management, the preparation of precise emergency plans, and the use of GIS methods to obtain these plans. Nevertheless, approaches of post-disaster accessibility analysis are not discussed.

**2 Case Study**

The case study fits the topic in general because of the described high exposure for earthquakes, the dense urban structure in the city core of Bucharest, the old and vulnerable building stock, and the described historical events. The description of the case study is detailed and comprehensive.

**3 Data and methods**

The section presents the used data sets and methods of the study. As suggested in the first review, the authors included a new subsection that focuses on the used data and their sources. What remains somewhat unclear is the consideration of travel modes. The described network also includes walking and cycling routes (see Line 190) that are unsuitable for emergency purposes in many cases. Please clarify this aspect.

Still, the methods section is not coherent (especially Lines 187–217). A lot of tools

and steps of analysis are mentioned (without detailed description or their background) without including their results in the following sections—e.g. different density estimations (only Kernel density was used later for visualisation purposes), assessment and estimation of potential damages, and exacerbation of impacts by gas stations. Please focus on the methods used to produce the results shown in the paper and provide details for step g) "determining, by simulation, the immediately inaccessible or poorly accessible areas" or assign the named tools in a structured manner.

The used method is limited because it is based on two assumptions: (1) Every building with the given classification (Risk I) collapses during the notional event and (2) every collapsed building leads to a road congestion and therefore to a functional loss of the street segment (although it should be represented by a function of building characteristics like age, number of storeys, material, and the surrounding space).

**4   Results and Discussion**

The section presents the results of the network analysis. The start of the section focuses on accessibilities before and after an earthquake event. Based on these analyses, recommendations for local authorities and spatial planners are given.

Figures 3, 4, and 5 all show the distribution of the building stock with a high seismic risk. I recommend excluding Figure 3 because the visualisation is very hard to read and not necessary when there is Figure 5. Figure 4 additionally shows traffic density—a linear representation of traffic based on the street network would be more reasonable. A big improvement would be the homogenization of Figure 4 and Figure 5 regarding the scale (Figure 4 seems to be clinched and a scale bar with 1320 m is very uncommon) and the extent of the maps to make them

comparable. Please also reconsider the design of Figure 5; the map has too many classes, there is no intuitive colouring, and density measures are missing in the legend.

Some analytical steps (connection between traffic and buildings at risk, the intersection of densities of buildings at risk and seismicity, wooden construction as fire sources) remain on a descriptive and basic level.

Figures 6 and 7 have been improved, but they should also focus on the historical centre/study area. In the presented visualisations, the historical centre shows a rather homogeneous accessibility and the details described in the text (Lines 285–296) are missing in the figures. Homogenisation with extent and scale of Figures 8 and 9 is highly recommended to make differences between pre- and post-event situations visible.

**5 General notes**

A separation of results and discussion may improve accessibility. There is a significant change of perspective at Line 312. The application of network analysis for mapping accessibilities is common sense. The methodological approach of the study utilizes buildings with seismic risks as potential road blocks in case of an earthquake event. Although the used approach is well-established and not new, the authors draw interesting recommendations and the results may be helpful for local authorities (there has been a noticeable improvement in this section). To further develop the method, the application of different scenarios (based on earthquake intensity or building indicators) is recommended for future work. Also, the discussion on limitations, validity, and accuracy of the methods and results remains open.

[Figure]

Although the authors have appreciably improved their paper, the described shortcomings should be carefully addressed before the material can become acceptable for final publication. Please also review the storyline of the article again. Therefore, I recommend **major revisions**.

---

## Author Comment (AC3) · 20 Jun 2018

Detailed responses to both Referees, and to S. Boengiu (SC)

First of all, we thank the Anonymous Referees and S. Boengiu for their careful analysis of our paper, and for the constructive suggestions on eliminating confusions, increasing the visibility of the results, and improving our proposed article.

Our detailed responses to them following:

**REFEREE 1**

1. Referee comments:

[Figure]

Compared to the previous version, the references improved. However, technical correction to Crowley, H., Colombi, M., Pinho, R., Meroni, F., and Cassera, A.: Application of a prioritisation scheme for seismic intervention in school buildings in Italy, in: 14th World Conf. Earthq. Eng., Beijing, China. Although the WCEE papers are archived in the web, there is a better referenceable paper by the authors in Earthquake Spectra (Damian N. Grant, Julian J. Bommer, Rui Pinho, G. Michele Calvi, Agostino Goretti, and Fabrizio Meroni (2007) A Prioritization Scheme for Seismic Intervention in School Buildings in Italy. Earthquake Spectra: May 2007, Vol. 23, No. 2, pp. 291-314.)

1. Authors' response:

It's right, we agree your recommendation seeing that the article published in Earthquake Spectra is better as reference, and the content is the same.

By consequence, we replaced, in the main text, at the line 53, Crowley et al., 2008, with Grant et al., 2007.

At the same time at the REFERENCES we made the same change, replacing:

Crowley, H., Colombi, M., Pinho, R., Meroni, F., and Cassera, A.: Application of a prioritisation scheme for seismic intervention in school buildings in Italy, in: 14th World Conf. Earthq. Eng., Beijing, China. ftp://ftp.ecn.purdue.edu/spujol/Andres/files/09-01-0097.PDF, Oct. 12-17, 2008,

with

Grant, D.N., Bommer, J.J., Pinho, R., Michele Calvi, G., Goretti, A., and Meroni, F. (2007) A Prioritization Scheme for Seismic Intervention in School Buildings in Italy. Earthq. Spectra, 23, 291-314, 2007. https://doi.org/10.1193/1.2722784

2. Referee comments:

The Frank Fiedrich article I suggested within the same collaboration is for example http://ieeexplore.ieee.org/abstract/document/4117644/?reload=true or

https://www.informs-sim.org/wsc06papers/059.pdf which included simulation of post-earthquake fire for Magheru Boulevard in Bucharest.

2. Authors' Response: Thank you for this recommendation! Indeed the paper of Frank Fiedrich completes our references giving, at the same time, the possibility to add new ideas connected with our topic.

Following to this, the authors added new phrases inside of the main document:

a) At the line 60, we add the following phrase: As Fiedrich (2007) suggests are fundamental the response actions to a disaster during the first three days after that, when the main goal is to fire fighting (if it is the case), to rescue the trapped victims, and to apply the urgency treatment of injured persons.

b) At the line 289, we introduce a new phrase: There are some studies on fire fighting simulation outside of historical center of Bucharest, in the Magheru Blvd (for example), which releave the importance given to this related phenomenon with an earthquake event (Fiedrich, 2007).

At the REFERENCES, we added:

Fiedrich, F.: An HLA-Based Multiagent System for Optimized Resource Allocation After Strong Earthquakes, Simulation Conference, 3-6 Dec.,WSC 06, Proc. Winter, Monterrey, CA, USA, 2006, added to IEEE Xplore: 05 March 2007, DOI: 10.1109/WSC.2006.323120

3. Referee comments:

More recent writings adressing urban infrastructure such as roads by the author are Urban Disaster Resilience and Security. Addressing Risks in Societies. Editors Alexander Fekete Frank Fiedrich (Springer).

3. Authors' response: Your suggestion apdated our literature on the topic, as we have found in this publication new confirmations of the importance of the accessibility in the

case of an earthquake event. By consequences we make some interventions in our paper:

At the line 236, we have added: a) In the recent years, the scientific approaches on risk reduction of natural events, as earthquakes use resilience, as an important concept, which could offer new theoretical and practical tools for a better civil protection (Fekete and Fiedrich, 2018). Using this concept, the scientists pave the way for revigoration the expectations, by joint actions with decision-makers and people (Anhorn, 2018). These ideas ask, maybe, other complementary issues connected with a higher accessibility to the affected areas.

We added to REFERENCES:

Anhorn, J.: Nepal and the "Urban Resilience Utopia", in Editors: Fekete A., Fiedrich F (eds), 2018, Urban Disaster Resilience and Security, pp. 13-26. The Urban Book Series, Springer, 2018. https://doi.org/10.1007/978-3-319-68606-6.

Fekete, A. and Fiedrich, F.: Introduction to "Urban Disaster Resilience and Security – Adressing Risks in Societies", in Editors: Fekete A., Fiedrich F (eds), 2018, Urban Disaster Resilience and Security, pp.1-12. The Urban Book Series, Springer, 2018. https://doi.org/10.1007/978-3-319-68606-6.

At the line 32, we have added: b) In a disaster situation, one of the most important elements is the public-private emergency cooperation, which can act in all the disaster phases. Developing a model to harmonise the joint cooperation, Wiens et al. (2018) identify some efficient ways to improve the logistics operations during the crisis management.

We added to REFERENCES:

Wiens, M., Schatter, F., Zobel C.W. and Schultmann, F. in Editors: Fekete A., Fiedrich F (eds), 2018, Urban Disaster Resilience and Security, pp.145-168. The Urban Book Series, Springer, 2018. https://doi.org/10.1007/978-3-319-68606-6.

4. Referee comments: Connected with the previous suggestion, the Referee 1 recommend another "recent writings": ………... Einführung in den Bevölkerungsschutz. Autoren: Fiedrich, Frank, Kud- lacek, Dominic (Springer)".

4. Authors' response:

This interesting book is not yeat printed! Having only some general information it was difficult for authors to use some ideas. Any case, in the following our approaches we will use it.

**REFEREE 2**

1. Referee comments:

"Nevertheless, approaches of post-disaster accessibility analysis are not discussed"

1. Authors' response:

At lines 33 (a), 54 (b), 66 (c), and 83 (d) we have added the followings:

a) In any disaster situation, one of the most important factors across all the disaster phases is public-private emergency cooperation for post-disaster accessibility and efficient intervention. By developing a model to harmonise this strong cooperation, Wiens et al. (2018) identify efficient ways to improve the logistics of these operations during crisis management.

b) Post-disaster recovery needs to transfer the most debated academic concepts (as disaster resilience, for example) into appropriate politics and transform it into real tools for an adequate planning. The governments have an important task to prepare the population and all stakeholders for future similar events (Comerio, 2014).

c) As Fiedrich (2007) suggests, the disaster responses made during the first three days are fundamental. After that, the main goals are invariably rescuing trapped victims, and treatment of the injured, though ongoing fire control may also be required in some cases.

d) A similar study, based on different hazard scenarios and a deep analysis on social vulnerability in Bucharest, identifies the importance of fire stations, hospitals and parks in post-disaster situations (ArmaÈŹ et al., 2016).

2. Referee comments:

"What remains somewhat unclear is the consideration of travel modes. The described network also includes walking and cycling routes (see Line 190) that are unsuitable for emergency purposes in many cases. Please clarify this aspect."

2. Authors' response: We have clarified the text in line 197 to show that, while walking and cycling networks have been considered in other studies, given their nature they have not figured in our research.

"The calculation of accessibility was initially based on the geometric structure of the public transport network (busses, trams and underground services), but not on the walking and cycling networks, which, although they have been included in other studies, are less amenable to emergency service access in this context (Graeme & Aylward, 1999; Parker & Campbell, 1998; Naphtali, 2006; Svensson, 2010; Weiping & Chi, 2011; Sotoudehnia & Comber, 2011; ESPON TRACC Interim Report, 2013; ESPON GROSSE, 2013; Blandford et al., 2012; Coffee et al., 2012; Yiannakoulias et al., 2013; Vojnovic et al., 2014).

3. Referee comments:

Still, the methods section is not coherent (especially Lines 187–217). A lot of tools and steps of analysis are mentioned (without detailed description or their background) without including their results in the following sections for e.g. different density estimations (only Kernel density was used later for visualisation purposes), assessment and estimation of potential damages, and exacerbation of impacts by gas stations. Please focus on the methods used to produce the results shown in the paper and provide details for step g) "determining, by simulation, the immediately inaccessible or poorly

accessible areas" or assign the named tools in a structured manner.

3. Authors' response: We have provided further details on the GIS technologies in the Methods section and provided further information on the techniques and the analysis in the Results and Discussion sections. Our paper focuses exclusively on the magnitude of the accessibility challenges resulting from the potential collapse of buildings in various risk categories, rather than dealing more generally with the evaluation and estimation of damage.

In our opinion detail for step g) is provided by Fig.10 and the text related to it in which we elaborate the hypothesis that, should an earthquake having the same intensity of the March 4, 1977 event occur, there is the potential for all the buildings in the Risk 1 category to collapse. Furthermore, it only requires building collapses sufficient to create one blocked street segment to achieve a decline in accessibility for civil protection. Fig.10 therefore indicates the areas and street segments that could be considered as potentially inaccessible or poorly accessible. Our intent here is to postulate and discuss an intuitive simulation (without applying formal simulation techniques), in order to demonstrate the importance of accessibility in this context.

At the same time, we have removed the following text from line 217 (it was a remanent phrase, from other very early version):

Accessibility was also calculated to take into account the presence of specific service locations which could exacerbate the impact of potential disasters, such as gas stations and electric transformers (Rezaie & Panahi, 2015).

4. Referee comments:

The used method is limited because it is based on two assumptions: (1) Every building with the given classification (Risk I) collapses during the notional event and (2) every collapsed building leads to a road congestion and therefore to a functional loss of the street segment (although it should be represented by a function of building characteristics like age, number of storeys, material, and the surrounding space).

4. Authors' response:

Our (in our opinion, defensible) position here is that the buildings classified in the Risk 1 category are more likely to collapse and thereby to cause accessibility challenges in the historical center of Bucharest in the case of a very strong earthquake. Axiomatically any building which flanks a street has the potential to collapse and impede accessibility. Certainly, it is possible that all the buildings in Category 1 may not collapse while some buildings outside this category may do so. This does not alter the fact that the greatest likelihood of building collapse, and therefore of street section blockage, will be where the concentrations of Risk 1 buildings are greatest.

5. Referee comments:

I recommend excluding Figure 3 because the visualisation is very hard to read and not necessary when there is Figure 5.

5. Authors' response:

We agree! Figure 3 has been removed (line 265).

6. Referee comments:

A big improvement would be the homogenization of Figure 4 and Figure 5 regarding the scale (Figure 4 seems to be clinched and a scale bar with 1320 m is very uncommon) and the extent of the maps to make them comparable. Please also reconsider the design of Figure 5; the map has too many classes, there is no intuitive colouring, and density measures are missing in the legend.

6. Authors' response:

We agree! The figures are now at the same scale and the number of classes in the Figure 5 has been reduced (lines 273 and 283)

7. Referee comments:

Figures 6 and 7 have been improved, but they should also focus on the historical centre/study area. In the presented visualisations, the historical centre shows a rather homogeneous accessibility and the details described in the text (Lines 285–296) are missing in the figures. Homogenisation with extent and scale of Figures 8 and 9 is highly recommended to make differences between pre- and post-event situations visible.

7. Authors' response:

We agree! Both maps have been replaced by others which are now at the same scale and clearly focus on the study area (lines 304 and 312).

8. Referee comments: A separation of results and discussion may improve accessibility. There is a significant change of perspective at Line 312.

8. Authors' response:

We agree with the referee's argument. Indeed, the original version of our paper incuded these two sections. We have therefore reverted to our initial structure as per the referee's recommendation, albeit with a slight modification in that we commence the discussion at line 315.

S. BOENGIU – SHORT COMMENTS

1. S. Boengiu comments:

The avoiding of the comment to the Fig.2b, respectively: "the photograph shows efforts to identify victims and property"

1. Authors' response:

We have removed "the photograph shows efforts to identify victims and property", and the explanation of the Fig.2b is:

"The collapse of the Continental block in Bucharest's historical centre in 1977 blocked the access streets, so clearance was delayed by more than 12 hours" (line 128)

2. S. Boengiu comments:

The streets canevas on the Fig.10 is not clear. Please, make clearer, for a better orientation of the readers2

2. Authors' response:

We have improved the Fig.10 (line 411), making the street network more visible.

3. S. Boengiu comments:

For uniformity of English, please replace in the entire manuscript all centre(s) with center(s), or inversely. See the followings lines: 120, 165, 166, 168 (inside the table), 204, 340, 352 and 412

3. Authors' response:

The spelling of "center" has been changed to "centre" as requested in the specified locations.

Please also note the supplement to this comment:
https://www.nat-hazards-earth-syst-sci-discuss.net/nhess-2018-41/nhess-2018-41-AC3-supplement.pdf

---

## Author Comment (AC4) · 20 Jun 2018

Detailed responses to both Referees, and to S. Boengiu (SC)

First of all, we thank the Anonymous Referees and S. Boengiu for their careful analysis of our paper, and for the constructive suggestions on eliminating confusions, increasing the visibility of the results, and improving our proposed article.

Our detailed responses to them following:

**REFEREE 1**

1. Referee comments:

[Figure]

Compared to the previous version, the references improved. However, technical correction to Crowley, H., Colombi, M., Pinho, R., Meroni, F., and Cassera, A.: Application of a prioritisation scheme for seismic intervention in school buildings in Italy, in: 14th World Conf. Earthq. Eng., Beijing, China. Although the WCEE papers are archived in the web, there is a better referenceable paper by the authors in Earthquake Spectra (Damian N. Grant, Julian J. Bommer, Rui Pinho, G. Michele Calvi, Agostino Goretti, and Fabrizio Meroni (2007) A Prioritization Scheme for Seismic Intervention in School Buildings in Italy. Earthquake Spectra: May 2007, Vol. 23, No. 2, pp. 291-314.)

1. Authors' response:

It's right, we agree your recommendation seeing that the article published in Earthquake Spectra is better as reference, and the content is the same.

By consequence, we replaced, in the main text, at the line 53, Crowley et al., 2008, with Grant et al., 2007.

At the same time at the REFERENCES we made the same change, replacing:

Crowley, H., Colombi, M., Pinho, R., Meroni, F., and Cassera, A.: Application of a prioritisation scheme for seismic intervention in school buildings in Italy, in: 14th World Conf. Earthq. Eng., Beijing, China. ftp://ftp.ecn.purdue.edu/spujol/Andres/files/09-01-0097.PDF, Oct. 12-17, 2008,

with

Grant, D.N., Bommer, J.J., Pinho, R., Michele Calvi, G., Goretti, A., and Meroni, F. (2007) A Prioritization Scheme for Seismic Intervention in School Buildings in Italy. Earthq. Spectra, 23, 291-314, 2007. https://doi.org/10.1193/1.2722784

2. Referee comments:

The Frank Fiedrich article I suggested within the same collaboration is for example http://ieeexplore.ieee.org/abstract/document/4117644/?reload=true or

https://www.informs-sim.org/wsc06papers/059.pdf which included simulation of post-earthquake fire for Magheru Boulevard in Bucharest.

2. Authors' Response: Thank you for this recommendation! Indeed the paper of Frank Fiedrich completes our references giving, at the same time, the possibility to add new ideas connected with our topic.

Following to this, the authors added new phrases inside of the main document:

a) At the line 60, we add the following phrase: As Fiedrich (2007) suggests are fundamental the response actions to a disaster during the first three days after that, when the main goal is to fire fighting (if it is the case), to rescue the trapped victims, and to apply the urgency treatment of injured persons.

b) At the line 289, we introduce a new phrase: There are some studies on fire fighting simulation outside of historical center of Bucharest, in the Magheru Blvd (for example), which releave the importance given to this related phenomenon with an earthquake event (Fiedrich, 2007).

At the REFERENCES, we added:

Fiedrich, F.: An HLA-Based Multiagent System for Optimized Resource Allocation After Strong Earthquakes, Simulation Conference, 3-6 Dec.,WSC 06, Proc. Winter, Monterrey, CA, USA, 2006, added to IEEE Xplore: 05 March 2007, DOI: 10.1109/WSC.2006.323120

3. Referee comments:

More recent writings adressing urban infrastructure such as roads by the author are Urban Disaster Resilience and Security. Addressing Risks in Societies. Editors Alexander Fekete Frank Fiedrich (Springer).

3. Authors' response: Your suggestion apdated our literature on the topic, as we have found in this publication new confirmations of the importance of the accessibility in the

case of an earthquake event. By consequences we make some interventions in our paper:

At the line 236, we have added: a) In the recent years, the scientific approaches on risk reduction of natural events, as earthquakes use resilience, as an important concept, which could offer new theoretical and practical tools for a better civil protection (Fekete and Fiedrich, 2018). Using this concept, the scientists pave the way for revigoration the expectations, by joint actions with decision-makers and people (Anhorn, 2018). These ideas ask, maybe, other complementary issues connected with a higher accessibility to the affected areas.

We added to REFERENCES:

Anhorn, J.: Nepal and the "Urban Resilience Utopia", in Editors: Fekete A., Fiedrich F (eds), 2018, Urban Disaster Resilience and Security, pp. 13-26. The Urban Book Series, Springer, 2018. https://doi.org/10.1007/978-3-319-68606-6.

Fekete, A. and Fiedrich, F.: Introduction to "Urban Disaster Resilience and Security – Adressing Risks in Societies", in Editors: Fekete A., Fiedrich F (eds), 2018, Urban Disaster Resilience and Security, pp.1-12. The Urban Book Series, Springer, 2018. https://doi.org/10.1007/978-3-319-68606-6.

At the line 32, we have added: b) In a disaster situation, one of the most important elements is the public-private emergency cooperation, which can act in all the disaster phases. Developing a model to harmonise the joint cooperation, Wiens et al. (2018) identify some efficient ways to improve the logistics operations during the crisis management.

We added to REFERENCES:

Wiens, M., Schatter, F., Zobel C.W. and Schultmann, F. in Editors: Fekete A., Fiedrich F (eds), 2018, Urban Disaster Resilience and Security, pp.145-168. The Urban Book Series, Springer, 2018. https://doi.org/10.1007/978-3-319-68606-6.

[Figure]

4. Referee comments: Connected with the previous suggestion, the Referee 1 recommend another "recent writings": ……….. Einführung in den Bevölkerungsschutz. Autoren: Fiedrich, Frank, Kud- lacek, Dominic (Springer)".

4. Authors' response:

This interesting book is not yeat printed! Having only some general information it was difficult for authors to use some ideas. Any case, in the following our approaches we will use it.

**REFEREE 2**

1. Referee comments:

"Nevertheless, approaches of post-disaster accessibility analysis are not discussed"

1. Authors' response:

At lines 33 (a), 54 (b), 66 (c), and 83 (d) we have added the followings:

a) In any disaster situation, one of the most important factors across all the disaster phases is public-private emergency cooperation for post-disaster accessibility and efficient intervention. By developing a model to harmonise this strong cooperation, Wiens et al. (2018) identify efficient ways to improve the logistics of these operations during crisis management.

b) Post-disaster recovery needs to transfer the most debated academic concepts (as disaster resilience, for example) into appropriate politics and transform it into real tools for an adequate planning. The governments have an important task to prepare the population and all stakeholders for future similar events (Comerio, 2014).

c) As Fiedrich (2007) suggests, the disaster responses made during the first three days are fundamental. After that, the main goals are invariably rescuing trapped victims, and treatment of the injured, though ongoing fire control may also be required in some cases.

d) A similar study, based on different hazard scenarios and a deep analysis on social vulnerability in Bucharest, identifies the importance of fire stations, hospitals and parks in post-disaster situations (ArmaÈŹ et al., 2016).

2. Referee comments:

"What remains somewhat unclear is the consideration of travel modes. The described network also includes walking and cycling routes (see Line 190) that are unsuitable for emergency purposes in many cases. Please clarify this aspect."

2. Authors' response: We have clarified the text in line 197 to show that, while walking and cycling networks have been considered in other studies, given their nature they have not figured in our research.

"The calculation of accessibility was initially based on the geometric structure of the public transport network (busses, trams and underground services), but not on the walking and cycling networks, which, although they have been included in other studies, are less amenable to emergency service access in this context (Graeme & Aylward, 1999; Parker & Campbell, 1998; Naphtali, 2006; Svensson, 2010; Weiping & Chi, 2011; Sotoudehnia & Comber, 2011; ESPON TRACC Interim Report, 2013; ESPON GROSSE, 2013; Blandford et al., 2012; Coffee et al., 2012; Yiannakoulias et al., 2013; Vojnovic et al., 2014).

3. Referee comments:

Still, the methods section is not coherent (especially Lines 187–217). A lot of tools and steps of analysis are mentioned (without detailed description or their background) without including their results in the following sections for e.g. different density estimations (only Kernel density was used later for visualisation purposes), assessment and estimation of potential damages, and exacerbation of impacts by gas stations. Please focus on the methods used to produce the results shown in the paper and provide details for step g) "determining, by simulation, the immediately inaccessible or poorly

accessible areas" or assign the named tools in a structured manner.

3. Authors' response: We have provided further details on the GIS technologies in the Methods section and provided further information on the techniques and the analysis in the Results and Discussion sections. Our paper focuses exclusively on the magnitude of the accessibility challenges resulting from the potential collapse of buildings in various risk categories, rather than dealing more generally with the evaluation and estimation of damage.

In our opinion detail for step g) is provided by Fig.10 and the text related to it in which we elaborate the hypothesis that, should an earthquake having the same intensity of the March 4, 1977 event occur, there is the potential for all the buildings in the Risk 1 category to collapse. Furthermore, it only requires building collapses sufficient to create one blocked street segment to achieve a decline in accessibility for civil protection. Fig.10 therefore indicates the areas and street segments that could be considered as potentially inaccessible or poorly accessible. Our intent here is to postulate and discuss an intuitive simulation (without applying formal simulation techniques), in order to demonstrate the importance of accessibility in this context.

At the same time, we have removed the following text from line 217 (it was a remanent phrase, from other very early version):

Accessibility was also calculated to take into account the presence of specific service locations which could exacerbate the impact of potential disasters, such as gas stations and electric transformers (Rezaie & Panahi, 2015).

4. Referee comments:

The used method is limited because it is based on two assumptions: (1) Every building with the given classification (Risk I) collapses during the notional event and (2) every collapsed building leads to a road congestion and therefore to a functional loss of the street segment (although it should be represented by a function of building characteristics like age, number of storeys, material, and the surrounding space).

4. Authors' response:

Our (in our opinion, defensible) position here is that the buildings classified in the Risk 1 category are more likely to collapse and thereby to cause accessibility challenges in the historical center of Bucharest in the case of a very strong earthquake. Axiomatically any building which flanks a street has the potential to collapse and impede accessibility. Certainly, it is possible that all the buildings in Category 1 may not collapse while some buildings outside this category may do so. This does not alter the fact that the greatest likelihood of building collapse, and therefore of street section blockage, will be where the concentrations of Risk 1 buildings are greatest.

5. Referee comments:

I recommend excluding Figure 3 because the visualisation is very hard to read and not necessary when there is Figure 5.

5. Authors' response:

We agree! Figure 3 has been removed (line 265).

6. Referee comments:

A big improvement would be the homogenization of Figure 4 and Figure 5 regarding the scale (Figure 4 seems to be clinched and a scale bar with 1320 m is very uncommon) and the extent of the maps to make them comparable. Please also reconsider the design of Figure 5; the map has too many classes, there is no intuitive colouring, and density measures are missing in the legend.

6. Authors' response:

We agree! The figures are now at the same scale and the number of classes in the Figure 5 has been reduced (lines 273 and 283)
7. Referee comments:

Figures 6 and 7 have been improved, but they should also focus on the historical centre/study area. In the presented visualisations, the historical centre shows a rather homogeneous accessibility and the details described in the text (Lines 285–296) are missing in the figures. Homogenisation with extent and scale of Figures 8 and 9 is highly recommended to make differences between pre- and post-event situations visible.

7. Authors' response:

We agree! Both maps have been replaced by others which are now at the same scale and clearly focus on the study area (lines 304 and 312).

8. Referee comments: A separation of results and discussion may improve accessibility. There is a significant change of perspective at Line 312.

8. Authors' response:

We agree with the referee's argument. Indeed, the original version of our paper included these two sections. We have therefore reverted to our initial structure as per the referee's recommendation, albeit with a slight modification in that we commence the discussion at line 315.

S. BOENGIU – SHORT COMMENTS

1. S. Boengiu comments:

The avoiding of the comment to the Fig.2b, respectively: "the photograph shows efforts to identify victims and property"

1. Authors' response:

We have removed "the photograph shows efforts to identify victims and property", and the explanation of the Fig.2b is:

"The collapse of the Continental block in Bucharest's historical centre in 1977 blocked the access streets, so clearance was delayed by more than 12 hours" (line 128)

2. S. Boengiu comments:

The streets canevas on the Fig.10 is not clear. Please, make clearer, for a better orientation of the readers2

2. Authors' response:

We have improved the Fig.10 (line 411), making the street network more visible.

3. S. Boengiu comments:

For uniformity of English, please replace in the entire manuscript all centre(s) with center(s), or inversely. See the followings lines: 120, 165, 166, 168 (inside the table), 204, 340, 352 and 412

3. Authors' response:

The spelling of "center" has been changed to "centre" as requested in the specified locations.

Please also note the supplement to this comment:
https://www.nat-hazards-earth-syst-sci-discuss.net/nhess-2018-41/nhess-2018-41-AC4-supplement.pdf

**Supplement:**

**Mapping Accessibility for Earthquake Hazard Response in the Historic Urban Centre of Bucharest**

Cristina Merciu[1], Ioan Ianoş[1], George-Laurenţiu Merciu[2], Roy Jones[3], and George Pomeroy[4]

[1]Interdisciplinary Centre of Advanced Research on Territorial Dynamics, University of Bucharest, Blvd. Regina Elisabeta, 4-12, code 030018, Romania
[2]Faculty of Geography, University of Bucharest, Blvd. Nicolae Bălcescu, 1, code 030018, Romania
[3]Geography Discipline Group, Curtin University, Perth, Western Australia 6845, Australia
[4]Geography – Earth Science Department, Shippensburg University of Pennsylvania. 1871 Old Main Drive, Shippensburg PA 17257

*Correspondence to*: Ioan Ianoş (ianos50@yahoo.com)

**Abstract.** Planning for post-disaster accessibility is essential for the provision of emergency and other services to protect life and property in impacted areas. Such planning is particularly important in congested historic districts where narrow streets and at-risk structures are more common and may even prevail. Indeed, a standard method of measuring accessibility, through the use of isochrones, may be particularly inappropriate in these congested historic areas. Bucharest, Romania, is a city with a core of historic buildings and narrow streets. Furthermore, Bucharest ranks second only to Istanbul among large European cities in terms of its seismic risk. This paper provides an accessibility simulation for central Bucharest using mapping and GIS technologies. It hypothesizes that all buildings in the Risk 1 class would collapse in an earthquake of a similar magnitude to those of 1940 and 1977. The authors then simulate accessibility impacts in the historic centre of Bucharest, such as the isolation of certain areas, and blockages of some street sections. In this simulation, accessibility will be substantially compromised by anticipated and extensive building collapse. Therefore, policy makers and planners need to fully understand and incorporate the serious implications of this compromised accessibility when planning emergency services and disaster recovery responses.

**1 Introduction**

A longitudinal analysis of natural hazards in major urban areas shows an increasing awareness of the frequency of disasters and especially of earthquakes (Eshghi & Larson, 2008; Armaş, 2012; Lu & Xu, 2014). Indeed, earthquakes are among the natural disasters that generate the greatest human and material losses (Geis, 2000;

30   Armaş & Avram, 2008; Atanasiu & Toma, 2012). Their impacts demand a prompt response from decision
     makers and the wider population, through the proper management of emergency situations (Waugh & Streib,
     2006). In any disaster situation, one of the most important factors across all the disaster phases is public-private
     emergency cooperation for post-disaster accessibility and efficient intervention. By developing a model to
     harmonise this strong cooperation, Wiens et al. (2018) identify efficient ways to improve the logistics of these
35   operations during crisis management.

     Many areas of high seismic risk are urbanized and densely populated (Pollino et al., 2012; Vatseva et al.,
     2013). In addition, and coincidentally, many countries experiencing economic transitions are characterized by
     urban growth that is uncontrolled and, in large and medium-sized urban centres, such growth can be especially
     chaotic (Salvati, 2014). Thus, an increase in the human and economic cost of such disasters can be reasonably
40   anticipated. Furthermore, many new buildings, new structures and, sometimes, newer pieces of infrastructure
     frequently fail to comply with the construction regulations established for areas of differing seismic
     vulnerability, especially when there are strong pressures for rapid development. Finally, the characteristically
     long time lags between pairs of strong earthquakes (Schweier & Markus, 2009) can dull public awareness of the
     potential impacts of such disasters, and render those in charge of emergency management complacent.

45   Earthquakes require a specific disaster planning approach (Armaş, 2008; Boştenaru Dan & Armaş, 2015).
     This is because, unlike disasters that can be anticipated in the short term (such as storms), there is little or no
     delay between the occurrence of the earthquake and the subsequent loss of life and property damage. Therefore,
     emergency response activities must be executed very quickly and efficiently (Wegscheider et al., 2013). For
     cities with a high earthquake risk, an important factor is public awareness of such events. This conditions the
50   population towards the importance of quick response measures, which can help to reduce property damage and,
     more importantly, the number of casualties (Armaş & Avram, 2008). However, no matter how well organized
     the mitigation process, the disastrous effects of major earthquakes cannot be totally avoided (Momani & Salmi,
     2012).

     Post-disaster recovery needs to transfer the most debated academic concepts (as disaster resilience, for
55   example) into appropriate politics and transform it into real tools for an adequate planning. The governments
     have an important task to prepare the population and all stakeholders for future similar events (Comerio, 2014).

     In recent years, seismic risk management has been more fully studied and developed so as to establish a series
     of priorities related to the rehabilitation of those buildings considered to be of major importance, including
     schools (Grant et al., 2007; Raffaelle et al., 2013; Panahi et al., 2013), public institutions, historic buildings, and
60   monuments (Grasso & Maugeri, 2009; Pessina & Meroni, 2009). Urban earthquake planning therefore needs to

**Comment [I.I.1]:** Authors' response to RC1 and RC2 (1)

**Comment [I.I.2]:** Adding a relevant issue – RC2 (1)

**Comment [I.I.3]:** Crowley at al., 2008 has been replaced by Grant et al., 2007, responding to suggestion of RC1

be more proactive (Boştenaru Dan et al., 2014) and there is a demonstrated requirement for coherent urban policies (Ianoş et al., 2017) to mitigate the inevitable occurrence of blockage points during emergency interventions.

In emergency situations, the key response element is rapid accessibility to places where possible casualties may be located. Timely intervention within the first two hours is critical in saving the wounded and in identifying the safest access routes for specific emergency equipment. As Fiedrich (2007) suggests, the disaster responses made during the first three days are fundamental. After that, the main goals are invariably rescuing trapped victims, and treatment of the injured, though ongoing fire control may also be required in some cases.

In general, natural hazard management includes the development of impact scenarios before the actual disasters occur (Bakillah et al., 2013). In this context, GIS techniques may be particularly useful in developing decision-making and response scenarios for potential earthquake disasters.

Our study shows that special attention should be paid to accessibility in the historic centres of large cities (Ianoş & Cepoiu, 2009). Historic city centres are characterized by intense pedestrian traffic and by a high proportion of attraction points (clubs, restaurants, hotels etc.) which result in high concentrations of people. Since the core of the historic centre of Bucharest is characterized by a high number of buildings that were strongly affected by earthquakes in the last century, we can reasonably speculate that determining their accessibility in an emergency situation will facilitate quick intervention in areas where injured people, either direct casualties or victims of earthquake-related phenomena such as fires, gas accumulations or local flooding, are likely to be concentrated. The main objective of the study is to integrate geospatial data using thematic mapping products with GIS techniques in order to provide seismic risk management solutions for Bucharest. We therefore seek to provide, concrete data and comprehensible information that can enable decision-makers to implement and prioritize their disaster management strategies. A similar study, based on different hazard scenarios and a deep analysis on social vulnerability in Bucharest, identifies the importance of fire stations, hospitals and parks in post-disaster situations (Armaş et al., 2016).

Unlike most studies of community response following an earthquake occurrence and the critical analysis of the emergency situations management generated thereby (Pollino et al., 2012; Wegscheider et al., 2013; Lu & Xu, 2014), the present study demonstrates the importance of GIS analyses in detecting potential congestion and inaccessibility issues in areas where buildings are most likely to collapse and accessibility issues are most likely to arise as a result of an earthquake.

**Comment [I.I.4]:** Authors' response to RC1 and RC2 (1)

**Comment [I.I.5]:** Adding a relevant reference - RC2 (1)

[revised manuscript text omitted]

**Comment [I.I.8]:** The phrase has been reformulated, answering to RC2 (2)

**Comment [I.I.9]:** The following phrase has been removed*:"Accessibility was also calculated to take into account the presence of specific service locations which could exacerbate the impact of potential disasters, such as gas stations and electric transformers (Rezaie & Panahi, 2015)"* (RC2 (3)

same time, GIS is a valuable method of analysis for this purpose because the databases can be regularly updated, allowing for ongoing mapping of the changing risk scenarios and the updating or reassessment of potential damage. The risk scenarios also provide useful identification of the vulnerable areas and population groups
225 (Sinha et al., 2008).

The penultimate methodological step was to identify likely congestion locations. The initial simulation assumed that all the buildings categorized as possessing the highest degree of risk would collapse. For the core of historical centre, this permitted the identification of some important sites and street segments, which would be blocked in the case of a strong earthquake using the location of each highest risk building, their age, and
230 number of floors and the local configuration of the street network.

Our intention is not to propose a precise correlation between the vulnerability of buildings (based on all their characteristics) and the intensity of the next earthquake. Rather, especially by taking into account that some of buildings in this area have partially collapsed in the absence of a direct seismic cause; we contend that an earthquake of similar magnitude to the 1977 event would produce outcomes comparable to our simulation. .
235 From this information several maps were developed taking into account the region's particular seismogenic characteristics (Mäntyniemi et al., 2003). Two offer general images of accessibility at the city level closely correlated with the territorial distribution of fire stations and hospitals. Another identifies areas or street segments potentially isolated by building collapses.

In the recent years, the scientific approaches on risk reduction of natural events, as earthquakes use resilience,
240 as an important concept, which could offer new theoretical and practical tools for a better civil protection (Fekete and Fiedrich, 2018). Using this concept, the scientists pave the way for revigoration the expectations, by joint actions with decision-makers and people (Anhorn, 2018). These ideas ask, maybe, other complementary issues connected with a higher accessibility to the affected areas.

Our approach, focusing on the single issue of accessibility in a situation of crisis management, shows
245 empirically how GIS technologies can be used to make recommendations to authorities to improve their preparedness levels and response speeds in post-earthquake interventions. Within this study, GIS is used solely as a tool to identify accessibility as a starting point for disaster management (Nushi & van Loenen, 2013). These GIS solutions are demonstrably important applications in relation to the first two phases (risk mitigation and disaster preparedness) of Alexander's (2002) four-phase sequence of emergency management activities,

**Comment [I.I.10]:** New added phrases at the suggestion of RC1 to update the literature on the topic

**4 Results**

**Comment [I.I.11]:** Splitting the "Results and discussion" section, we have individualised the "Results" and "Discussion" sections – RC2 (8)

250    It is necessary to simulate emergency interventions prior to the occurrence of catastrophic events because, in the local situation, the inherited intra-urban structure, with a narrow winding street pattern  dating back to medieval times, the poor structural condition of many of the buildings, and  limited access to important points from  the emergency response activity locations are all of critical importance.

[revised manuscript text omitted]

320 impossible.

**Comment [I.I.17]:** Splitting the "Results and discussion" section, we have individualised the "Results" and "Discussion" sections – RC2 (8)

[revised manuscript text omitted]

Comment [I.I.25]: New added reference according with suggestion of RC1

---

## Author Comment (AC6) · 20 Jun 2018

Thank you so much for your suggestions improving our paper. The changes have been made in the main manuscript.